# EmoFeedback²: Reinforcement of Continuous Emotional Image Generation via LVLM-based Reward and Textual Feedback

## Abstract

Continuous emotional image content generation (C-EICG) is emerging rapidly due to its ability to produce images aligned with both user descriptions and continuous emotional values. However, existing approaches lack emotional feedback from generated images, limiting the control of emotional continuity. Additionally, their simple alignment between emotions and naively generated texts fails to adaptively adjust emotional prompts according to image content, leading to insufficient emotional fidelity. To address these concerns, we propose a novel generation-understanding-feedback reinforcement paradigm (EmoFeedback²) for C-EICG, which exploits the reasoning capability of the fine-tuned large vision–language model (LVLM) to provide reward and textual feedback for generating high-quality images with continuous emotions. Specifically, we introduce an emotion-aware reward feedback strategy, where the LVLM evaluates the emotional values of generated images and computes the reward against target emotions, guiding the reinforcement fine-tuning of the generative model and enhancing the emotional continuity of images. Furthermore, we design a self-promotion textual feedback framework, in which the LVLM iteratively analyzes the emotional content of generated images and adaptively produces refinement suggestions for the next-round prompt, improving the emotional fidelity with fine-grained content. Extensive experimental results demonstrate that our approach effectively generates high-quality images with the desired emotions, outperforming existing state-of-the-art methods in our custom dataset. The code and dataset will be released soon.

## 1 Introduction

Emotions play a crucial role in shaping our perception and understanding of the world, deeply influencing how we interact with our environment (Chainay et al., 2012; Yang et al., 2018). Among the many stimuli that evoke emotions, visual cues stand out as particularly powerful due to their intuitiveness and richness of information. Researchers have extensively explored the field of Visual Emotion Analysis (VEA) (Borth et al., 2013; Megalakaki et al., 2019; Rao et al., 2020; Wang et al., 2022) to investigate the complex interplay between visual content and human emotions. In recent years, the rapid advancement of generative models (Ho et al., 2020; Rombach et al., 2022; Esser et al., 2024) has enabled them to produce visual content with impressive quality based on textual descriptions. In content creation, incorporating emotional elements is often more effective in engaging and resonating with audiences. However, studies on models capable of generating images reflecting specific emotions remain limited.

Current methods typically construct an emotion encoding network to derive emotional representations from text prompts and guide pre-trained generative models to produce images expressing corresponding emotions. EmoGen (Yang et al., 2024) pioneered emotion-driven image generation using discrete tags (e.g., happy, sad) via a mapping network, but was limited by categorical labels. To overcome this, EmotiCrafter (Dang et al., 2025) proposed Continuous Emotion Image Content Generation (C-EICG), embedding Valence and Arousal (Russell, 1980) into prompts for nuanced control. Complementing this, EmoEdit (Yang et al., 2025) introduced paired emotional–original datasets and an Emotion Adapter to align target emotions with visual inputs. However, all of these methods face several challenges: (1) Lack of emotional feedback from generated images: Their

training objective is to align the features extracted by the emotion encoder with the target semantics, while the actual emotions expressed in the generated images are not fed back to the model for optimization. As a result, the model fails to capture the subtle variation of emotions in images, constraining its ability to control emotional continuity. (2) Insufficient adaptability in addressing the affective gap (Zhao et al., 2021): Since users exhibit an affective gap in understanding emotional descriptions, texts are often absent in real-world testing, with only emotional values being provided. Existing methods inject emotions by aligning these values with simply generated texts, but lack the adaptability to flexibly adjust emotional prompts according to image content, leading to limited emotional fidelity.

To overcome the two limitations above, we propose EmoFeedback[2], a novel generation-understanding-feedback reinforcement paradigm to provide Large Vision Language Model (LVLM)-based reward and textual feedback for C-EICG. Figure 1 represents the overall framework of our method. Specifically, we introduce the Group Relative Policy Optimization (GRPO) (Shao et al., 2024) framework with multi-task to endow the Qwen2.5-VL-7B-Instruct (Bai et al., 2025) with emotion understanding ability. Subsequently, we present an emotion-aware reward feedback strategy to better capture the intrinsic relationship between visual content and emotional expression. The LVLM acts as a reward model to measure the discrepancy between predicted and target emotional values, providing emotional feedback to optimize the Stable Diffusion 3.5-Medium (SD3.5-M) (Esser et al., 2024) generative model and strengthen control over emotional continuity. In addition, we propose a self-promotion textual feedback optimization framework to adaptively generate the emotional prompts. In each iteration, SD3.5-M produces multiple candidate images, from which the most and least emotion-aligned samples are selected for comparative analysis. Leveraging its chain-of-thought reasoning capability, the LVLM can produce prompt refinement suggestions to enrich emotional descriptions and content details of the next-iteration prompt, thereby improving emotional fidelity and expressiveness.

To summarize, our main contribution can be listed as:

- We propose a novel generation-understanding-feedback reinforcement paradigm for C-EICG, exploiting the reasoning ability of a fine-tuned emotional LVLM to provide reward and textual feedback for high-quality and emotionally continuous image generation.

- We introduce an emotion-aware reward feedback strategy that leverages the LVLM to assess the emotions of generated images and deliver emotional reward to drive the reinforcement optimization, enabling continuous and precise emotional control.

- We design a self-promotion textual feedback framework to analyze the generated content and adaptively optimize the emotional prompts, enhancing emotional fidelity through iterative enrichment and refinement of content details.

- We construct a custom dataset based on EmoSet-118K, which includes images, corresponding textual descriptions, Valence and Arousal values, as well as emotion categories. Extensive experimental results show that our method outperforms existing techniques in our custom dataset.

## 2 RELATED WORKS

### 2.1 VISUAL EMOTION ANALYSIS

VEA aims to computationally recognize emotions in images and videos. Early work emphasized discrete categories (Yang et al., 2020; 2022; Xu et al., 2022), but the emerging continuous models highlight dimensions such as arousal, valence, and dominance (Kollias, 2022; Toisoul et al., 2021). Recent studies integrate contextual cues from posture, objects, and scenes (Kosti et al., 2017; Kragel et al., 2019), achieving strong performance. The central inquiry, what evokes visual emotions, has been explored through low-level features (color, texture) and high-level features (content, style), with contributions such as SentiBank (Borth et al., 2013) and MldrNet (Rao et al., 2020). These efforts lay the foundation for generative approaches that embed emotions directly into visual content.

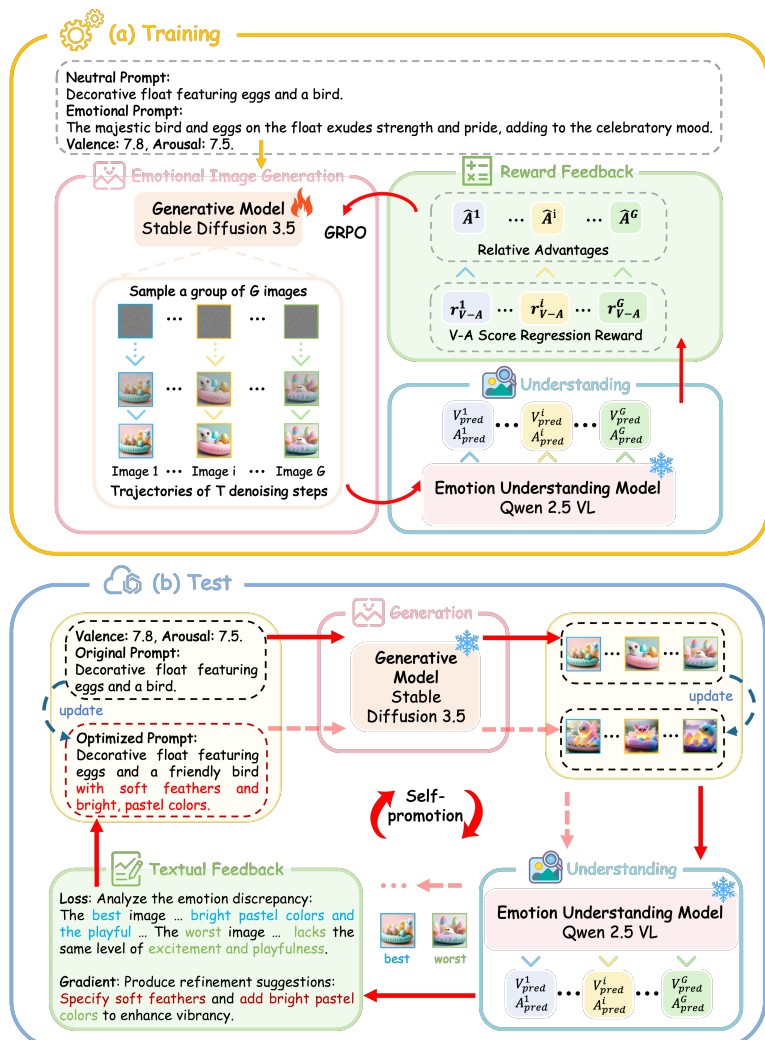

Figure 1: The framework of the EmoFeedback[2]. During training, given a neutral prompt, an emotional prompt, the V and A scores, the generative model produces a set of images. The emotion understanding model then evaluates the images to provide reward feedback. During testing, the emotional prompt is omitted due to the users' affective gap, and the model instead iteratively generates textual feedback to refine the prompts.

## 2.2 EMOTIONAL IMAGE GENERATION

Most of the previous works in EICG can be grouped into color-based (Chen et al., 2020; Liu et al., 2018; Yang & Peng, 2008; Peng et al., 2015), and style-based (Fu et al., 2022; Sun et al., 2023; Weng et al., 2023). Recently, EmoGen (Yang et al., 2024) pioneered the Emotion Image Content Generation task by generating images based on discrete emotion tags (e.g., happy, sad). The model presents a mapping network to transform abstract emotions into concrete concepts. While groundbreaking, this approach is restricted by the narrow scope of categorical emotion labels, which fail to capture nuanced affective states. To address this limitation, EmotiCrafter (Dang et al., 2025) introduced the C-EICG task, along with an emotion-embedding network that injects continuous Valence (V) and Arousal (A) values (Russell, 1980) into text prompts to enable smooth, emotion-driven image variation. Meanwhile, EmoEdit (Yang et al., 2025) constructed paired datasets of emotional and original images, designing an Emotion Adapter to mediate interactions between target emotions and input visuals. Different from previous works, our method incorporates emotional feedback from outputs to optimize the model and adaptively enrich emotional texts according to the image content.

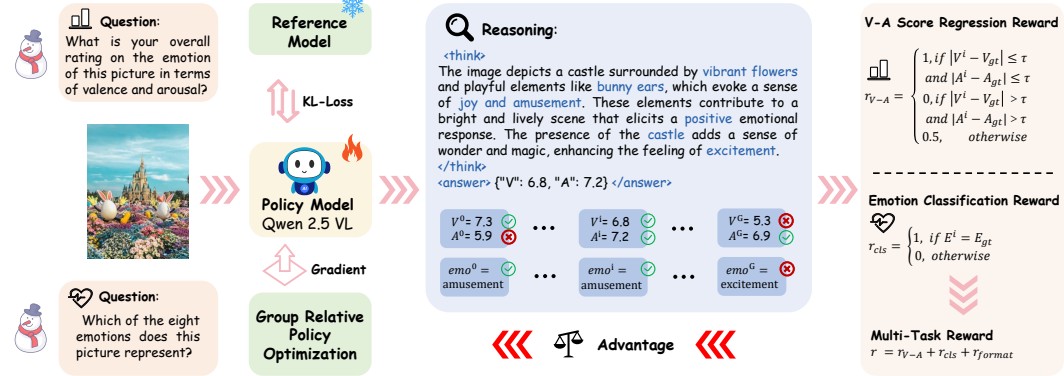

Figure 2: The Emotion Understanding Model Training Process. The training image is input into the emotion understanding model to predict the V-A scores and emotion labels. The set of outputs is then fed into the designed reward functions to calculate the reward. The GRPO algorithm finally derives the advantage and loss to optimize the policy model.

## 3 METHOD

### 3.1 EMOTION UNDERSTANDING MODEL TRAINING

For the emotion understanding model, we adopt GRPO for multi-task reinforcement fine-tuning of the Qwen2.5-VL-7B-Instruct model (Bai et al., 2025; Li et al., 2025a). In general, for each data pair, we design three reward functions to calculate the corresponding reward $r_i$ for each response $o_i$ generated by the policy model. Figure 2 demonstrates the training pipeline of the emotion understanding model. We describe the three reward functions as follows.

**Format Reward**: This reward enforces structured outputs: reasoning within "<think>" and "</think>" tags, answer within"<answer>" and "</answer>" tags. Moreover, the content inside "<answer>" must follow a JSON-like format (Guo et al., 2025). The reward score is set to 1 if the i-th response satisfies all the above conditions; otherwise, it is 0. [1]

**V–A Score Regression Reward**: This reward guides the model to reason about the degree of emotion expressed in the image along the two dimensions of emotion, i.e., valence and arousal. For each response $o_i$, the predicted values $(V^i, A^i)$ are compared with the ground-truth values $(V_{gt}, A_{gt})$. If the discrepancy falls within a predefined threshold $\tau$, a reward is assigned. This allows predictions with acceptable deviations to receive positive feedback without requiring exact matches. The formulation of reward $r_{V\text{-}A}$ is demonstrated in the Figure 2.

**Emotion Classification Reward**: This task encourages the model to accurately identify the discrete emotion category expressed in the image, facilitating the understanding of the emotional effect of certain objects. The model predicts one of eight emotion types: amusement, awe, anger, contentment, disgust, fear, excitement, and sadness. We design a binary reward $r_{cls}$ for this task: if the predicted emotion category $E^i$ matches the ground truth label $E_{gt}$, the reward is 1; otherwise, it is 0. The formulation of the reward $r_{cls}$ is demonstrated in Figure 2.

### 3.2 EMOTION-AWARE REWARD FEEDBACK

We design an emotion-aware reward feedback strategy for the reinforcement fine-tuning of SD3.5-M by employing the emotion understanding model to assess the generated images. Following the training paradigm of Flow-GRPO (Liu et al., 2025), for each text prompt, the model performs $T$ denoising steps and generates a group of $G$ images. Each image $\{x_0^i\}_{i=1}^G$ corresponds to a trajectory $\{(x_T^i, \ldots, x_0^i)\}_{i=1}^G$. Next, the emotion understanding model predicts V–A scores of each image and computes reward values as reward feedback. The group-level rewards are then normalized to compute the advantage of each image, thereby converting the absolute rewards into relative advantages

---
[1]The detailed prompts are provided in Appendix D.1.

that reflect the intra-group ranking:

$$\hat{A}^i = \frac{R\left(x_0^i, c\right) - \text{mean}\left(\left\{R\left(x_0^i, c\right)\right\}_{i=1}^G\right)}{\text{std}\left(\left\{R\left(x_0^i, c\right)\right\}_{i=1}^G\right)}. \tag{1}$$

For every step $t$ along the sampled trajectory, the model calculates the importance sampling weight $r_t^i$. To avoid overly large policy updates and ensure stable training, GRPO constrains $r_t^i$ within the range [1-$\delta$, 1+$\delta$]. In addition, a KL divergence penalty scaled by $\beta$ is applied to keep the learned policy close to the reference distribution $\pi_{\text{ref}}$. The final optimization objective of the generative model can be formulated as the expectation of weighted advantages across all samples and all timesteps within the group:

$$\mathcal{J}(\theta) = \mathbb{E}_{x^i \sim \pi_{\theta_{\text{old}}}} \frac{1}{G} \sum_{i=1}^G \frac{1}{T} \sum_{t=0}^{T-1} \left[ \min\left(r_t^i(\theta)\hat{A}^i, \; \text{clip}(r_t^i(\theta), 1-\varepsilon, 1+\varepsilon)\hat{A}^i\right) - \beta D_{\text{KL}}\left(\pi_\theta \parallel \pi_{\text{ref}}\right) \right], \tag{2}$$

where $r_t^i(\theta) = \pi_\theta(x_{t-1}^i|x_t^i, c)/\pi_{\theta_{\text{old}}}(x_{t-1}^i|x_t^i, c)$. Additionally, to mitigate reward hacking where the model overfits to emotional cues at the cost of severe content distortion, we incorporate the PickScore model, a human-preference-based metric, as an additional reward. Our generative model is therefore jointly optimized with both emotional fidelity and semantic consistency in the images.

### 3.3 Self-promotion Textual Feedback

In this work, we leverage the emotion understanding and chain-of-thought reasoning capabilities of LVLM to propose a self-promotion textual feedback framework. This framework adapts the core principle of gradient descent to the text-to-image generation (Yuksekgonul et al., 2024). Instead of updating model parameters with numerical losses and gradients, we treat the discrepancy between generated images and target emotions as the "loss", and the textual feedback provided by the LVLM as the "gradient", to improve the emotional quality of generated images during inference.

The textual feedback optimization consists of three key steps analogous to standard gradient optimization: loss computation, gradient estimation, and variable update. Formally, let $t$ denote the user prompt, $e$ the target emotion, $v$ the generated visual content, $\mathcal{M}$ the LVLM (Qwen2.5-VL-7B-Instruct), and $P$ the prompt function that specifies the instruction for each step. The optimization proceeds as follows:

**Loss computation**: The LVLM is instructed by $P_{\text{loss}}$ to evaluate the emotions of a group of generated images and compute the discrepancy from the target emotions as the loss:

$$\mathcal{L}(e, v) \leftarrow \mathcal{M}\left(P_{\text{loss}}(e, v)\right). \tag{3}$$

**Gradient estimation**: Guided by $P_{\text{grad}}$, the LVLM analyzes the images and their emotional loss, producing natural language suggestions for improving emotional quality:

$$\frac{\partial \mathcal{L}}{\partial v} \leftarrow \mathcal{M}\left(P_{\text{grad}}(\mathcal{L}(e, v))\right). \tag{4}$$

**Variable update**: Finally, under the instruction $P_{\text{update}}$, the LVLM refines the user prompt based on the gradient-like analysis and suggestions:

$$t_{\text{opt}} \leftarrow \mathcal{M}\left(P_{\text{update}}(\frac{\partial \mathcal{L}}{\partial v}, t)\right). \tag{5}$$

The optimized prompt $t_{\text{opt}}$ is then fed into the generative model $\mathcal{G}$ (SD3.5-M) in the next iteration to produce a new group of images. In practice, for each image group, we select the best and worst samples according to their losses, $\mathcal{L}_{\text{best}}$ and $\mathcal{L}_{\text{worst}}$, to the LVLM (Li et al., 2025b). By enriching the prompt with additional details and emotional cues, the newly generated images can better align with the desired emotions, while the model parameters remain fixed. [2]

---

[2]The generation–evaluation–feedback paradigm can be formalized with the pseudo code in Appendix B. The specific prompts to generate textual feedback could refer to Appendix D.1. The reasoning process and results of textual feedback could refer to Appendix E.2.

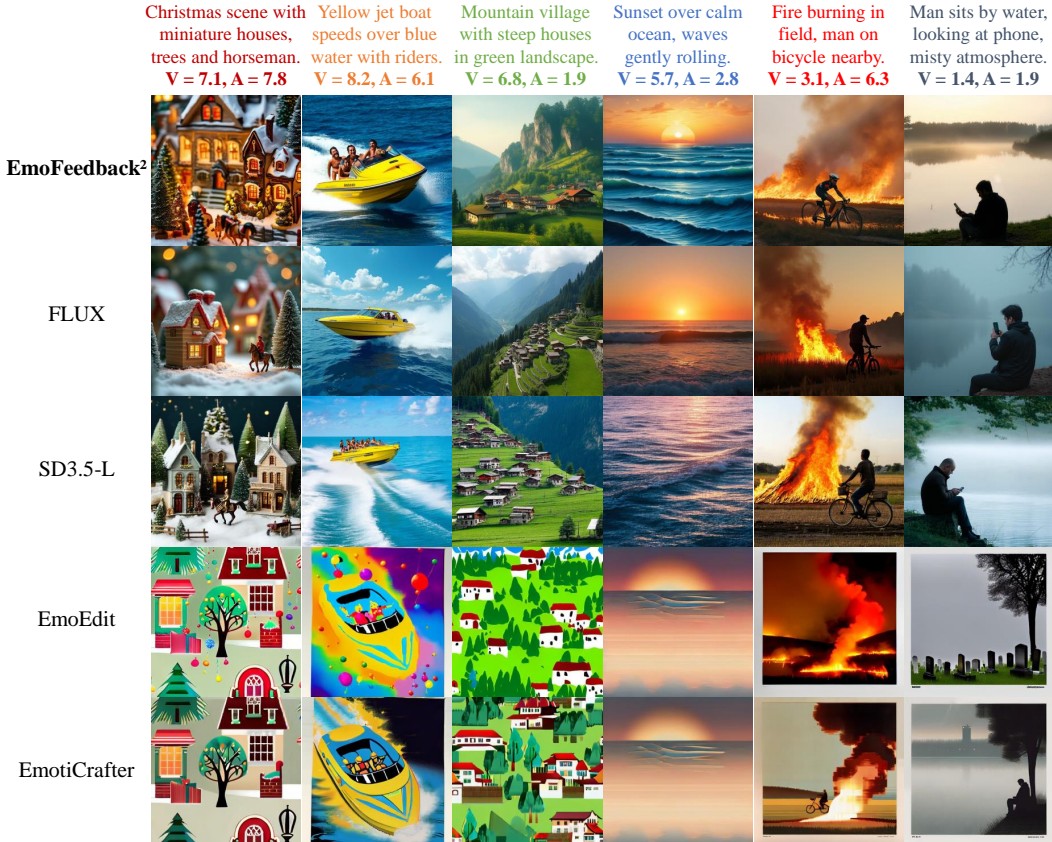

Figure 3: Qualitative comparisons with baselines under specific emotional states. Our approach demonstrates superior performance in many kinds of emotions.

## 4 EXPERIMENT

### 4.1 DATASET CONSTRUCTION

We construct a multimodal emotion dataset based on EmoSet-118K (Yang et al., 2023). The original EmoSet-118K only provides images and discrete emotion categories, lacking textual descriptions and V–A annotations. To address this limitation, we design a data construction pipeline. First, we employ a multimodal large language model (MLLM) to generate neutral, emotional prompts for each image in the training dataset, thereby obtaining image–text pairs. In the test dataset, we only generate the neutral prompts because in the real world, the emotional prompts are unavailable due to the affective gap. Next, following the emotion lexicon proposed in (Warriner et al., 2013), which maps emotion-related words to Valence and Arousal scores, we extract the mean and standard deviation of the V–A values for each emotion category. Based on these statistics, we build independent Gaussian distributions for each emotion label and randomly sample a pair of the V–A values from the corresponding distribution for each image. Through this process, we create a new dataset comprising 14,563 training samples and 1000 test samples.

### 4.2 BASELINE AND EVALUATION METRICS

To effectively evaluate the accuracy of our method in generating images that convey emotions, while also demonstrating the superior quality and aesthetics of the generated images, we selected four corresponding baselines for comparison. EmotiCrafter (Dang et al., 2025), EmoEdit (Yang et al., 2025): FLUX (Labs, 2024): the most powerful flow-matching method in Text2Image generation. Stable-Diffusion3.5-Large (SD3.5-L) (Esser et al., 2024): typical generative method.

We assess our method based on five metrics: V-Error, A-Error, CLIP-Score (Hessel et al., 2021), CLIP-IQA (Wang et al., 2023), and Aesthetic Score (Aes-Score). V-Error and A-Error evaluate the absolute error of predicted valence and arousal values and the target valence and arousal val-

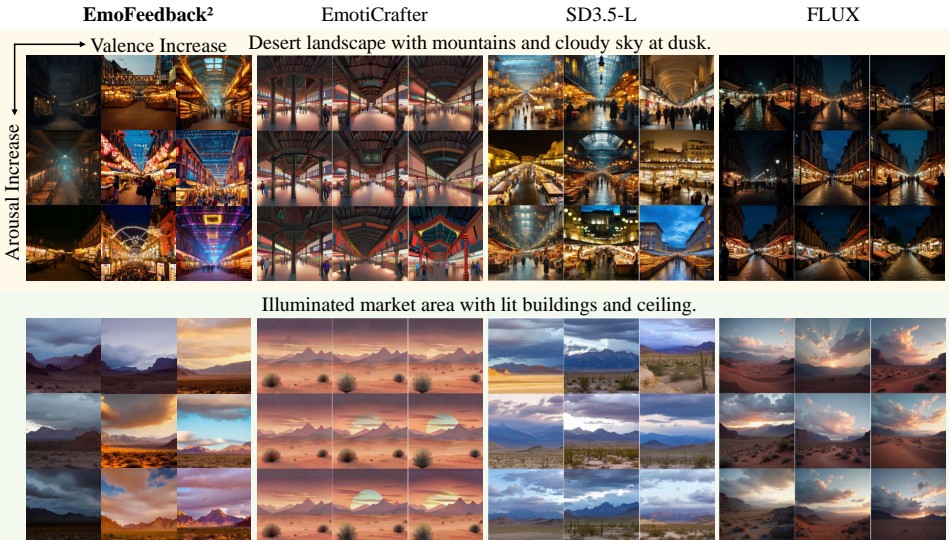

Figure 4: Qualitative comparisons with baselines under varying emotional values.

ues, representing the accuracy of emotional expression. CLIP-Score measures how well the image semantically aligns with a given text prompt, ensuring that the content matches the intended description. CLIP-IQA focuses on perceptual quality, assessing whether the image looks sharp, natural, and free of distortions or artifacts. Aesthetic Score reflects the subjective visual appeal, estimating how pleasing or artistically attractive the image is. [3]

### 4.3 COMPARISON

#### 4.3.1 QUALITATIVE COMPARISON

We evaluate the generated images based on three criteria: (1) the effectiveness of emotion expression, (2) the quality of the generated images, and (3) the continuity of visual changes as V and A values vary. The qualitative results are presented in Figure 3 and Figure 4.

Figure 3 demonstrates the performance of different methods in generating image content corresponding to specific emotional states. EmoFeedback[2] excels at preserving prompt content and effectively integrating emotional details. In contrast, EmotiCrafter struggles with conveying emotions in background content and has lower visual quality. EmoEdit often overemphasizes emotions, distorting the original image, like removing a cyclist or replacing a person with tombstones. SD3.5-L and FLUX generate high-quality visuals but fail to accurately depict emotions, with mismatched facial expressions or lacking energetic posture. [4]

Figure 4 illustrates how EmoFeedback[2] and baseline methods generate images that evolve with varying V-A values. Since EmoEdit can only take discrete emotion words as input, it fails to produce images under continuous shifts in V-A values. As shown, the image content generated by SD3.5-L and FLUX exhibits little perceptible change with V-A, as these models were not explicitly designed to capture emotional dynamics. EmotiCrafter demonstrates some degree of emotional changes, such as altering texture colors or adding sun elements at high V-A values. However, EmoFeedback[2] presents much more pronounced emotional expression in terms of background objects, brightness, color tone, and overall atmosphere, making it more effective in achieving emotionally coherent and visually compelling results. [5]

#### 4.3.2 QUANTITATIVE COMPARISON

As shown in Table 1, our method achieves the best (lowest) V-Error and A-Error on average, and also obtains the highest performance on CLIP-Score and CLIP-IQA, and suboptimal on Aes-Score,

---

[3]Refer to the Appendix B for detailed experiment settings.

[4]Refer to the Appendix E.2 for more qualitative experimental results.

[5]Refer to the Appendix E.2 for more qualitative experimental results.

Table 1: Performance comparison across various methods on the custom dataset. Best scores are in blue , second-best in green .

| Method | V-Error ↓ | A-Error ↓ | CLIP-Score ↑ | CLIP-IQA ↑ | Aes-Score ↑ |
|---|---|---|---|---|---|
| EmotiCrafter | 0.700 | 1.011 | 24.011 | 0.753 | 5.235 |
| EmoEdit | 0.545 | 0.927 | 21.932 | 0.800 | 5.228 |
| SD3.5-L | 0.552 | 0.753 | 25.209 | 0.834 | 5.335 |
| FLUX | 0.700 | 0.852 | 25.666 | 0.817 | 5.569 |
| EmoFeedback[2] | 0.521 | 0.710 | 26.889 | 0.880 | 5.442 |

Table 2: Performance comparison across various methods on the EMOTIC dataset. Best scores are in blue , second-best in green .

| Method | V-Error ↓ | A-Error ↓ | CLIP-Score ↑ | CLIP-IQA ↑ | Aes-Score ↑ |
|---|---|---|---|---|---|
| EmotiCrafter | 1.253 | 1.288 | 27.072 | 0.909 | 5.430 |
| SD3.5-L | 1.131 | 1.300 | 27.704 | 0.930 | 5.302 |
| FLUX | 1.047 | 1.395 | 27.877 | 0.930 | 5.597 |
| EmoFeedback[2] | 0.849 | 0.669 | 27.410 | 0.938 | 5.480 |

demonstrating both the accuracy of our generated images in conveying emotions and their superior image quality. For EmoEdit, it achieves the suboptimal result in V-Error at the cost of the lowest CLIP-Score, as it modifies images to inject emotional representation. For EmotiCrafter, in cases such as background images, where emotional expression is inherently ambiguous, it struggles to generate images that accurately convey the intended emotions, and its performance is further constrained by the quality of the prompts. Moreover, we can observe that after reinforcement fine-tuning and test-time text feedback optimization, our generative model surpasses the FLUX and SD3.5-L model in image quality and image-text alignment, highlighting the effectiveness of our approach.

Furthermore, to validate the generalization ability of our method, we conduct an extra validation experiment on the cross-domain dataset, EMOTIC (Kosti et al., 2019). EMOTIC provides continuous V-A annotations, it fundamentally differs from our custom dataset: EMOTIC's annotations describe the emotions of people in the image (e.g., facial expressions, body posture), whereas our dataset focuses on the global emotional expression of the overall content and scene. Thus, the emotion distribution in EMOTIC represents an unseen domain for our model. We conducted an additional evaluation on the EMOTIC test set. The emotional image generation results of EmoFeedback[2] and the baselines are included in the Table 2. EmoFeedback[2] achieves SOTA performance in V-Error, A-Error, and CLIP-IQA, along with a competitive CLIP-Score. These results demonstrate that our method maintains superior performance in terms of both emotional fidelity and image quality even in OOD scenarios, highlighting a significant generalization advantage over current SOTA methods.

### 4.3.3 USER STUDY

We conducted the user study to evaluate user preference for our proposed method. A total of 20 participants were recruited. The study assessed users' preferences in terms of emotional expressiveness and image quality. It consisted of 30 image sets, each containing five images generated by our model, and four baseline models. For each image set, participants were presented with the five images and asked two questions: (1) Which of the five images best conveys the target emotion? (2) Please rate each image on a scale of 1 to 5 (1 = poor, 5 = excellent) in terms of visual quality.

Table 3 presents the results of user preference for emotional expressiveness and average image quality ratings across the five methods. EmoFeedback[2] achieves a preference rate of 53.83%, significantly outperforming all four baselines. The results demonstrate that our method generates emotionally compelling content that aligns closely with human perception. Additionally, EmoFeedback[2] achieves the highest score of 4.00, surpassing all competing methods and confirming its superior visual fidelity and generation stability.

Table 3: User study comparison across 20 subjects. Best scores are in blue

| Metric | EmotiCrafter | EmoEdit | SD3.5-L | FLUX | EmoFeedback[2] |
|---|---|---|---|---|---|
| Emotional Preference | 2.00% | 4.00% | 18.33% | 21.83% | 53.83% |
| Image Quality | 1.89 | 1.92 | 3.79 | 3.85 | 4.00 |

Table 4: Ablation Study between different reward functions and model size. Best scores are in blue

| Model | Ours | Qwen-3B-S | Qwen-7B-C |
|---|---|---|---|
| V-Error | 0.521 | 0.628 | 0.819 |
| A-Error | 0.710 | 1.217 | 0.896 |

Table 5: Ablation Study between multi-task and single-task. Best scores are in blue

| Task | Jointly Training | Regression Only | Classification Only |
|---|---|---|---|
| V-Error | 0.521 | 0.579 | 1.445 |
| A-Error | 0.710 | 0.812 | 2.073 |

Table 6: Comparison between Lexicon-based Annotation and Human Annotation.

| Annotator | Lexicon-based Annotation | Human Annotation |
|---|---|---|
| V-Error | 0.521 | 0.781 |
| A-Error | 0.710 | 1.310 |

## 4.4 ABLATION STUDY

### 4.4.1 EMOTION UNDERSTANDING MODEL

Our emotion understanding model is based on the Qwen2.5-VL-7B-Instruct backbone, trained with a step reward function and a multi-task strategy. In this experiment, we first evaluate how the LVLM size and reward function design affect emotion assessment accuracy. Specifically, we replace the backbone with Qwen2.5-VL-3B-Instruct (Qwen-3B-S) to study the model size, and substitute the step reward with a continuous function of V–A discrepancy (Qwen-7B-C) to study the reward design. As shown in Table 4, our 7B model outperforms Qwen-3B-S on both metrics, and the step reward yields better performance across metrics than the continuous reward variant. Additionally, we validate the effectiveness of multi-task training. Table 5 shows that our jointly trained multi-task model significantly outperforms single-task regression-only and classification-only baselines, demonstrating that classification helps improve V-A score regression accuracy.

### 4.4.2 REWARD AND TEXTUAL FEEDBACK

We qualitatively evaluate the impact of reward feedback (RF) and textual feedback (TF) in EmoFeedback[2] on the emotional content of the generated images, as shown in Figure 5. The first row of images represents those generated using only the SD3.5-M model. The second and third rows show the results generated with reward feedback and textual feedback, respectively. The last row presents the results using both feedbacks. We can conclude that reward feedback enables the image to have emotional content from the initial generation, while textual feedback primarily enriches the details of the generated image to improve emotional expressiveness.

### 4.4.3 HUMAN ANNOTATION RESULTS

To validate our reliability of the lexicon-based annotations, we invite an additional eight experts to independently rate Valence and Arousal (1–9 scale) on 1,000 test images. As shown in Figure 6, the sorted distributions of lexicon-based and human annotations exhibit highly consistent trends, confirming that our method preserves the global structure of emotional perception. Furthermore, the comparable V-Error and A-Error rates reported in Table 6 demonstrate that our model is effectively aligned with human perception.

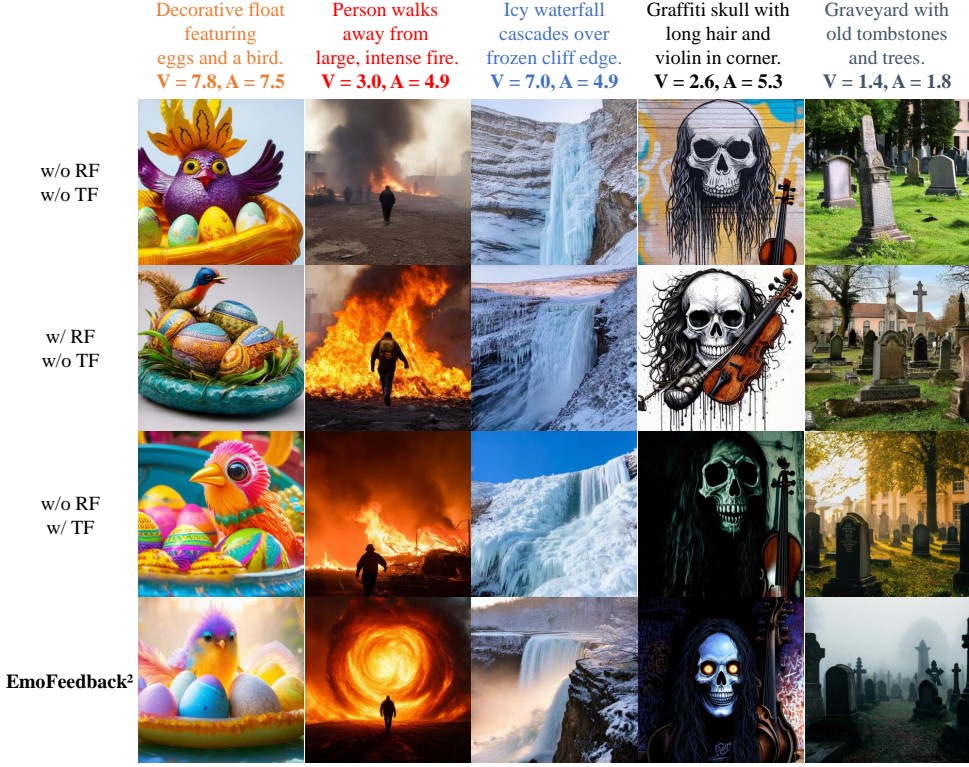

Figure 5: Ablation study on the reward and textual feedback.

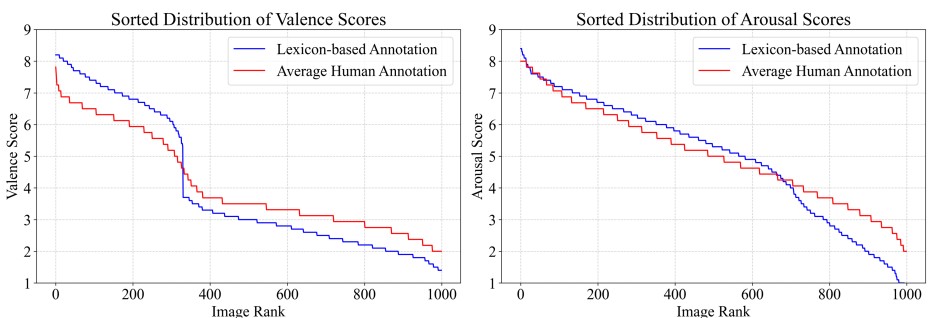

Figure 6: Additional Ablation Study on the valence distribution difference between human annotation and lexicon-based annotation

## 5 CONCLUSION

In this paper, we introduce EmoFeedback[2], a novel LVLM-based generation-understanding-feedback reinforcement paradigm for continuous emotional image content generation. Based on the emotion understanding LVLM, we introduce an emotion-aware reward feedback strategy, in which the LVLM calculates emotional reward feedback from generated images to fine-tune the generative model. At the inference stage, we further propose a self-promotion textual feedback optimization framework to adaptively provide refined emotional prompts for the next-round generation. Extensive experiments prove the effectiveness of our method: qualitative experiments show that EmoFeedback[2] generates emotionally expressive images whose content varies smoothly with changes in V–A values, while quantitative evaluation demonstrates that our method achieves state-of-the-art emotional fidelity and image quality, outperforming both emotional image content generation and general Text2Image generation fields. However, our current method lacks exploration of process reward models (PRM). The model's steps and elements for evaluating image emotion are still unclear. Additionally, since emotion understanding is highly subjective and each person's emotions are specific, the current unified training paradigm will have deviation for each user. Future work will focus on PRM and calibrating data according to each user's emotional preference.

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

APPENDIX

## A  THE USE OF LARGE LANGUAGE MODELS

In this work, we use Large Language Models (LLMs) to aid and polish writing. We utilize LLMs to refine the quality of our manuscript by suggesting more precise terminology. Additionally, we use LLMs to optimize LaTeX templates for figures, tables, and mathematical expressions, significantly reducing the time and effort required for typesetting complex layouts.

## B  FURTHER DETAILS ON THE EXPERIMENT SETUP

### B.1  MODELS AND BASELINE METHODS

The following table lists the generation model, the emotion understanding model, and the baseline methods.

| Models | Links |
| --- | --- |
| SD3.5-M | https://huggingface.co/stabilityai/stable-diffusion-3.5-medium |
| Qwen2.5-VL-7B-Instruct | https://huggingface.co/Qwen/Qwen2.5-VL-7B-Instruct |
| SD3.5-L | https://huggingface.co/stabilityai/stable-diffusion-3.5-large |
| FLUX.1-dev | https://huggingface.co/black-forest-labs/FLUX.1-dev |
| EmotiCrafter | https://github.com/idvxlab/EmotiCrafter |
| EmoEdit | https://github.com/JingyuanYY/EmoEdit |

### B.2  HYPERPARAMETERS SPECIFICATION

In the training of the emotion understanding model, the GRPO generation number $N$ is set to 8, the batch size is set to 16, and we train 5 epochs for convergence. The weight of the KL divergence penalty $\beta$ is set to 1e-3, while the weights $\alpha_1$ and $\alpha_2$ are set to 0.25 and 0.75, respectively. The threshold $\epsilon$ is set to 0.70. We employ AdamW as the optimizer, using an initial learning rate of 1e-6 that linearly decays to 1e-9 during training. In the training of the generation model, the GRPO generation number $N$ is set to 8. We use a sampling timestep $T = 10$ and an evaluation timestep $T = 25$. The image resolution is 512, and the KL ratio is set to 0.1. We set the training process to 1000 steps, and the batch size of every step is set to 16. During the self-promotion textual feedback framework, we set the iteration number of feedback to 3, and generate 8 images every iteration.

### B.3  SELF-PROMOTION TEXTUAL FEEDBACK FRAMEWORK

---
**Algorithm 1** Self-Promotion Textual Feedback Framework

---
**Require:**
    original prompt $t_0$, target emotion $e$, generative model $\mathcal{G}(\cdot)$
    max iterations $I$, prompt functions $P_{\text{loss}}, P_{\text{grad}}, P_{\text{update}}$
**Ensure:** optimized image $v_I$
1: $v_0 \leftarrow \mathcal{G}(t_0)$                                                     ▷ initial image
2: **for** $i = 0$ **to** $I - 1$ **do**
3:     $\mathcal{L}(e, v_i) \leftarrow \mathcal{M}\big(P_{\text{loss}}(e, v_i)\big)$                           ▷ textual loss
4:     $\mathcal{L}_{\text{best}}, \mathcal{L}_{\text{worst}} \leftarrow \text{select-best-worst}\big(\mathcal{L}(e, v_i)\big)$
5:     $\partial\mathcal{L}/\partial v_i \leftarrow \mathcal{M}\big(P_{\text{grad}}(\mathcal{L}_{\text{best}}, \mathcal{L}_{\text{worst}})\big)$           ▷ textual gradient
6:     $t_{i+1} \leftarrow \mathcal{M}\big(P_{\text{update}}(\partial\mathcal{L}/\partial v_i, t_i)\big)$             ▷ prompt update
7:     $v_{i+1} \leftarrow \mathcal{G}(t_{i+1})$                                   ▷ new image
8: **end for**
9: **return** $v_I$

---

## C    COMPUTATION EFFICIENCY

Our self-promotion textual feedback framework provides configurable iterative refinement at test time. Users can choose to perform a single generation round or request additional refinement. In each round, the model produces a batch of images, allowing the user to either pick a satisfactory result immediately or wait for the next refinement step, depending on their preference for emotional fidelity. Since our model is fine-tuned with emotion-aware reward feedback, it's capable of generating competitive emotional images even in a single round. Therefore, although our method supports iterative generation, the actual inference time and resource consumption depend entirely on the user's preference. For example, on a single H20 GPU, generating 8 images per round takes 11s, while the LVLM-based emotion evaluation and prompt refinement take an additional 9s. This means the minimal latency is 11s if the user selects images from the first round, and users who desire higher emotional fidelity can wait for another 20s, and so on. Therefore, generating 8 samples with three iterations needs 51s. We compare the average inference latency per sample between our model and the baselines, and the results are reported in Table 7. Our inference latency is comparable to that of current models. This improvement is large because our base generator, SD3.5-M, contains only 2.5B parameters, making it much lighter than other generative models. Therefore, our approach maintains efficient inference while offering superior emotional controllability, making it potentially practical for real-world deployment.

Table 7: Comparison of inference latency per sample and base generator model size.

| Metric | EmotiCrafter | SD3.5-L | FLUX | EmoFeedback[2] |
|---|---|---|---|---|
| Inference Latency (s) | 1.5 | 4.0 | 17.0 | 6.4 |
| Model Size (B) | 3.5 | 8.0 | 12.0 | 2.5 |

## D    LVLM'S EMOTIONAL EVALUATION MECHANISM

### D.1    EXPLICIT PROMPT GUIDANCE

As shown in Table 8, 9, the prompts used during training and evaluation explicitly instruct the model to focus on visual elements highly relevant to emotion, such as: "Please consider the weather, light, background object, and facial expression in the decision." The multimodal alignment mechanism of LVLMs enables the model to prioritize these features in visual encoding, which are known to be important in human emotional perception. The explicit prompts can introduce inductive bias that significantly influences the distribution of attention weights in LVLMs. Therefore, by leveraging the model's strong instruction-following capability, we improve the controllability and interpretability of emotion assessment.

### D.2    CHAIN-OF-THOUGHT REASONING

Utilizing the reasoning capabilities and hierarchical feature extraction of LVLM to reveal the process of emotional attribution and cue integration. We instruct the model to explicitly output its reasoning steps, making emotion judgment no longer implicitly encoded but expressed through an interpretable reasoning path. Figure 2 provides a concrete example. Before outputting the final emotion scores, the model clearly states its reasoning: (1) identifying key visual elements such as "castle," "flowers," and "bunny ears"; (2) interpreting visual attributes such as "bright" referring to color, "playful" referring to style, and "surrounded" referring to composition; (3) linking these elements to emotional implications, such as "amusement," "positive," "excitement". Through hierarchical feature extraction, the LVLM simultaneously captures low-level visual cues and high-level element attributes, and uses cross-modal associations to map visual features into an abstract emotional semantic space. This forms a coherent reasoning process that greatly enhances the reliability and interpretability of our method.

Table 8: **Prompts for Different Tasks.** The system prompt is shared across all tasks, while task-specific prompts are additionally designed for each task.

**System Prompt:** A conversation between User and Assistant. The user asks a question, and the Assistant solves it. The assistant first thinks about the reasoning process in the mind and then provides the user with the answer. The reasoning process and answer are enclosed within <think> </think> and <answer> </answer> tags, respectively, i.e., <think> reasoning process here </think><answer> answer here </answer>.

**Prompt for VA-Value Regression Task:** What is your overall rating on the valence and arousal of this picture? The valence and arousal rating should be a float between 1 and 9, rounded to two decimal places. For valence, 1 represents very sad and 9 represents very happy. For arousal, 1 represents very calm and 9 represents very active. Please consider the weather, light, background object, and facial expression in the decision. Return the result in JSON format with the following keys: "valence": The evaluated valence score. and "arousal": The evaluated arousal score.

**Prompt for Emotion Classification Task:** Analyze the given image and decide which of the following eight emotions the image represents: "amusement", "anger", "awe", "contentment", "fear", "disgust", "excitement", and "sadness". Please consider the weather, light, background objects, and facial expression in the decision. Return the result in JSON format with the following keys: "emotion_class": The detected emotion (or "null" if none).

# E EXTENDED EXPERIMENTAL RESULTS

## E.1 ADDITIONAL ABLATION STUDY

### E.1.1 REWARD HACKING

"Reward Hacking" manifests when the model sacrifices image quality or semantic content to maximize the emotion reward (e.g., generating a solid red patch to represent "anger"). PickScore is a CLIP-based scoring model trained on extensive human preference data, designed to evaluate both image-text alignment and aesthetics. We selected it specifically for its ability to penalize content distortion. Suppose the model distorts the original prompt in pursuit of emotional expression (e.g., a "dog" no longer resembles a dog). In that case, PickScore assigns a significantly lower score, thereby offsetting the high Emotion Reward. This mechanism forces the model to strike a balance between emotional fidelity and semantic consistency/aesthetic quality. To demonstrate the necessity of PickScore, we conducted an ablation study by training a variant of EmoFeedback² that excludes the PickScore reward. We selected sample images generated by both models at various training stages for comparison. In our experiment, PickScore is incorporated as an additional reward term. Within our reinforcement learning (GRPO) framework, we compute a combined reward defined as the weighted sum of the emotion fidelity reward. As shown in Figure 7, the results indicate that images generated by the model trained without PickScore gradually lose semantic features as training progresses. Conversely, the model trained with the multi-reward objective maintains semantic consistency and image quality.

### E.1.2 COMPARISON TO GENERAL LVLM-BASED MODEL

Most existing feedback frameworks rely on generic LVLMs. In contrast, a core contribution of our work is the development of an emotion-understanding LVLM (Section 3.1). We fine-tune it using GRPO with multi-task rewards (V-A regression reward and emotion classification reward). This expert LVLM evaluates emotional content more accurately and more consistently than a general-purpose LVLM. To quantitatively demonstrate the advantage of our expert LVLM and generation-understanding-feedback paradigm, we additionally constructed a general LVLM-based feedback baseline. This baseline utilizes a general Qwen2.5-VL-7B-Instruct model without our emotion-specific fine-tuning to provide feedback. Furthermore, we add the human preference reward model, like ImageReward and PickScore. The comparative results are included in Table 10. As the general LVLM lacks emotion-understanding capabilities, its V-Error (0.850) and A-Error (0.842) are significantly worse than those of our method (V-Error 0.521, A-Error 0.710). While ImageReward and PickScore show improved V-A performance compared with the general LVLM, our

Table 9: **Prompts for self-promotion and VA-Change.** The system prompt is shared across all tasks, while specific prompts are additionally designed for each individual task.

**System Prompt:** You are an expert in image emotion evaluation. You should first think about the reasoning process in your mind and then provide the user with the answer. The emotion metrics to assess images are Valence (V) and Arousal (A): Valence measures how positive or negative the emotion evoked by the image is. A score of 1 indicates extremely negative emotion, while 9 indicates extremely positive emotion. Arousal (A) measures how calming or stimulating the image is. A score of 1 indicates very calm or passive, while 9 indicates very exciting or active. The V and A rating should be two float values between 1 and 9. You will be given a text prompt containing target Valence and Arousal values, along with two images generated from this prompt by a diffusion model, and their corresponding evaluated emotional scores.

**Prompt for self-promotion textual feedback** 1. Analysis: Compare the best image with the worst image. Based on the relationship and differences between their evaluated emotional values and the target values, analyze the strengths and weaknesses of each image. Also, analyze and specify what aspects of the best image need to be modified or improved to make its emotional values closer to the target. Consider aspects such as lighting and brightness, weather and environment, color and composition, characters and objects in your analysis. 2. Optimization: To achieve a new image that aligns more closely with the target emotional values, rewrite and optimize the original text prompt with a more detailed emotional description. The optimized prompt should guide the diffusion model to generate an image that incorporates the modifications and improvements identified in the analysis. The optimized prompt must be richer in content than the original prompt. It should introduce meaningful modifications in aspects such as lighting, brightness, weather, environment, colors, composition, characters, or objects, while preserving the core semantics of the original prompt. 3. Return the answer only as a valid JSON object with exactly two keys: $analysis$: a single string with the comparative analysis, $optimized\_prompt$: the new optimized prompt as a string, in short word format. Do not include any explanations, headings, or Markdown, only return raw JSON.

**Prompt for VA-Change task:** 1. Analysis: Given target valence (V) $gt\_v$ and arousal (A) $gt\_a$, for instance, Low V, Low A to gloomy, depressive; Low V, High A to tense, fearful; High V, Low A to calm, serene; High V, High A to joyful, energetic. Describe the atmosphere the image should present. Provide guidance under four aspects: Lighting and brightness, Weather and environment, Color and composition, Characters and objects. 2. Optimization: Based on the analysis, rewrite and enrich the original text prompt so it better reflects the target emotions $gt\_v$, $gt\_a$. Use more detailed emotional description, considering typical V/A mappings: 3. Return only a valid JSON object with two keys: $analysis$: the analysis as one string, $optimized\_prompt$: the improved prompt as one string. Do not include any explanations, headings, or Markdown; only return raw JSON.

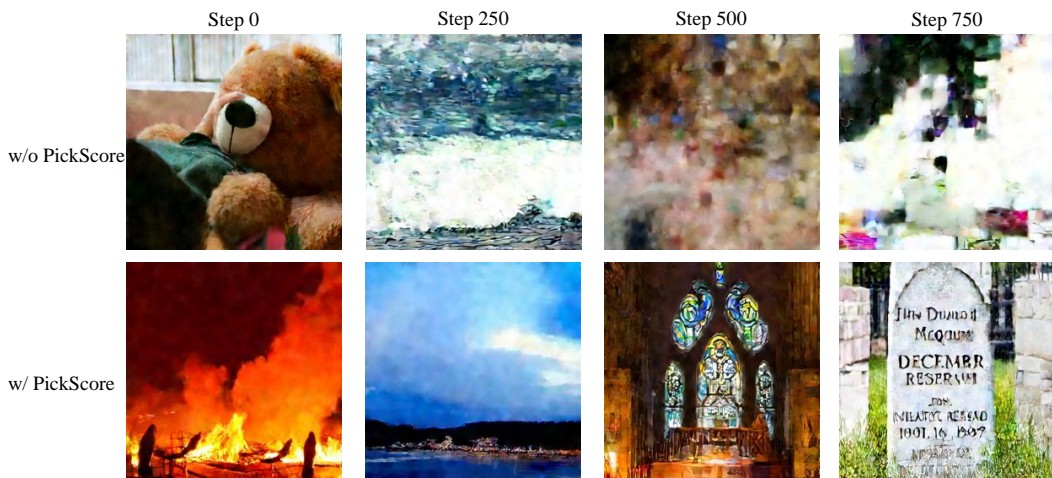

Figure 7: Additional Ablation Study results under reward hacking.

Table 10: Performance comparison across different reward models. Best scores are in blue, second-best in green.

| Method | V-Error ↓ | A-Error ↓ | CLIP-Score ↑ | CLIP-IQA ↑ | Aes-Score ↑ |
|---|---|---|---|---|---|
| Pickscore | 0.554 | 0.788 | 25.515 | 0.790 | 5.337 |
| ImageReward | 0.584 | 0.825 | 24.933 | 0.723 | 5.326 |
| General | 0.850 | 0.842 | 26.776 | 0.821 | 5.399 |
| Emotion | 0.521 | 0.710 | 26.889 | 0.880 | 5.442 |

method still outperforms them in all metrics. The results indicate that human preference may implicitly add some emotional information in the reward model. However, their performance remains inferior to the explicit emotion-aware modeling provided by our expert LVLM.

## E.2 ADDITIONAL QUALITATIVE RESULTS

Figures 8, 9, 10 qualitatively compare EmoFeedback[2] and other baselines. Figures 11, 12, 13 demonstrate the content variation of the pictures with the change of V and A values in different original emotions.

## E.3 THE TEXTUAL FEEDBACK

Table 11 and Table 12 represent two examples of the reasoning process in our proposed self-promotion textual feedback framework.

## F BROADER IMPACT

Our paradigm EmoFeedBack[2], enables the personalized generation of emotionally evocative images tailored to individual users and holds substantial promise. Leveraging the reasoning and feedback capabilities of Large Vision-Language Models (LVLMs) together with user preference data enables the development of private, user-specific models for emotional image generation. Furthermore, emotional valence-arousal (V-A) features can be decoded directly from electroencephalography (EEG) signals, opening up the possibility for users to enrich images with additional emotional content based on their neural responses to original visual stimuli.

Table 11: Examples of the self-promotion textual feedback, including analysis and outputs.

**Original:** Decorative float featuring eggs and a bird-like creature.

**Iteration 1:** Analysis: The best image has a bright and cheerful color palette, which gives it a positive and lively feeling. The bird-like creature adds a whimsical element that enhances the overall visual appeal. The worst image, on the other hand, has a darker and more intense color scheme, making it feel less appealing and more unsettling. The best image should be slightly more dynamic and vibrant to match the target emotional values.

Outputs: Decorative float featuring eggs and a friendly bird-like creature with soft feathers and bright, pastel colors.

**Iteration 2:** Analysis: The best image has a higher valence and arousal compared to the worst image. The bright pastel colors and the playful presence of the bird-like creature contribute to a more vibrant and stimulating scene. The worst image, despite having a soft and warm feeling, lacks the same level of excitement and playfulness.

Outputs: Decorative float featuring eggs and a friendly bird-like creature with soft feathers and bright, pastel colors, glowing under a soft morning light with a slightly hazy background.

Table 12: Examples of the self-promotion textual feedback, including analysis and outputs.

**Original:** Person walks away from large, intense fire.

**Iteration 1:** Analysis: The best image depicts a person walking away from a large, intense fire, which evokes a strong sense of danger and urgency. The evaluated emotional values align well with the target emotions. The worst image shows a person running towards the fire, which adds a dynamic element but may evoke a slightly different emotional response due to the direction of movement. The best image could benefit from a slight increase in the Arousal value to better match the target emotional values.

Outputs: Person walks away from a massive, swirling inferno amidst a dark, smoky night sky.

**Iteration 2:** Analysis: The best image has a slightly lower Arousal value compared to the target, making it more suitable for the given prompt. The worst image has a higher Arousal value, which may not align well with the intended feeling of the prompt.

Outputs: Person walks away from a massive, swirling inferno amidst a dark, smoky night sky with a glowing, fiery aura enveloping the scene.

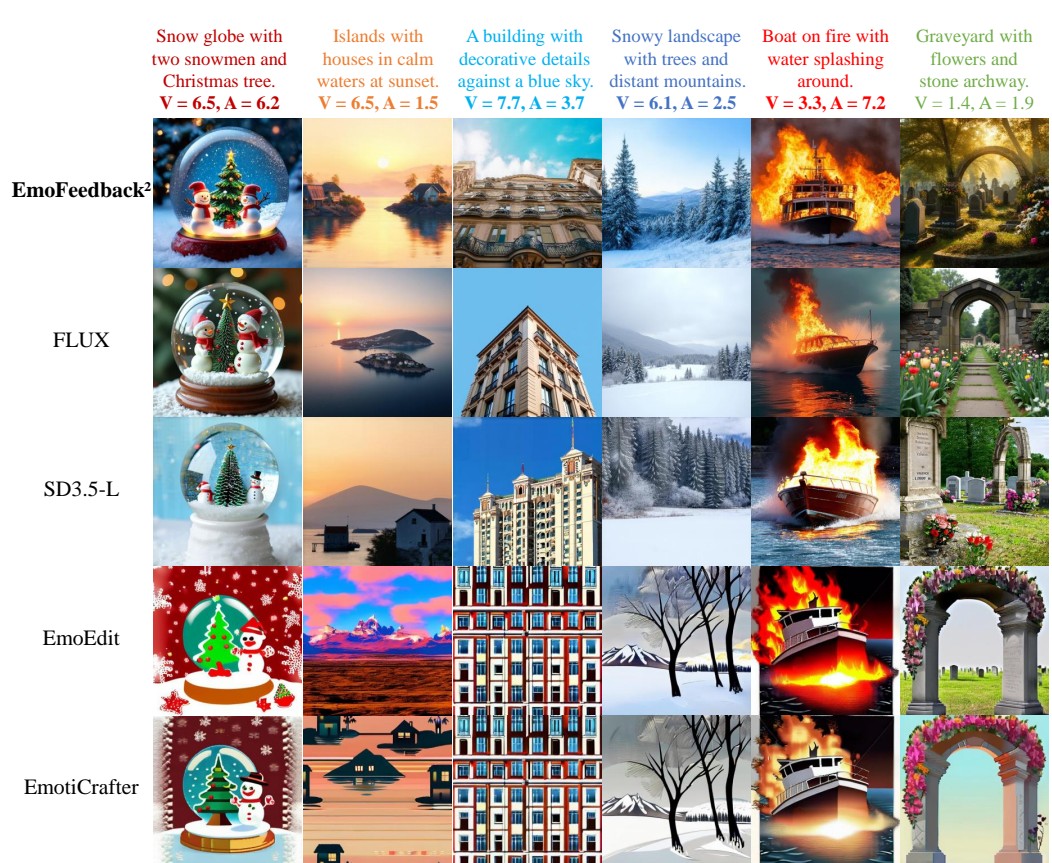

Figure 8: Additional Qualitative results under specific emotional states.

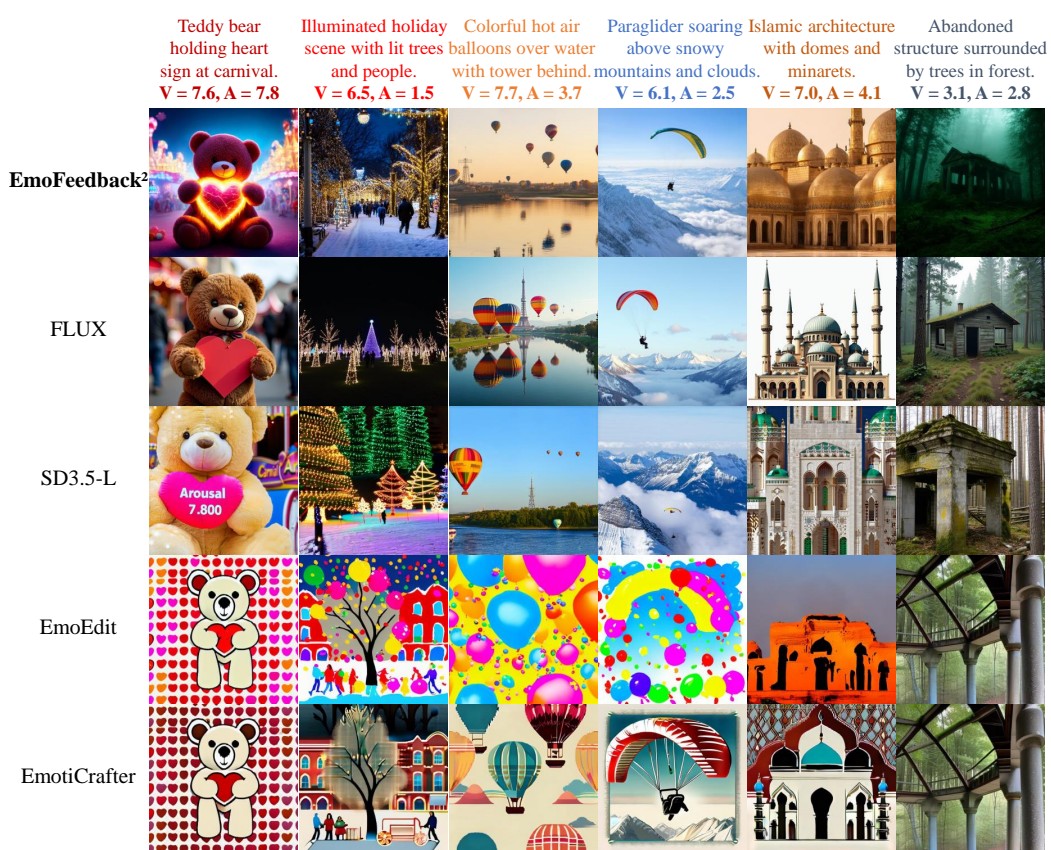

Figure 9: Additional Qualitative results under specific emotional states.

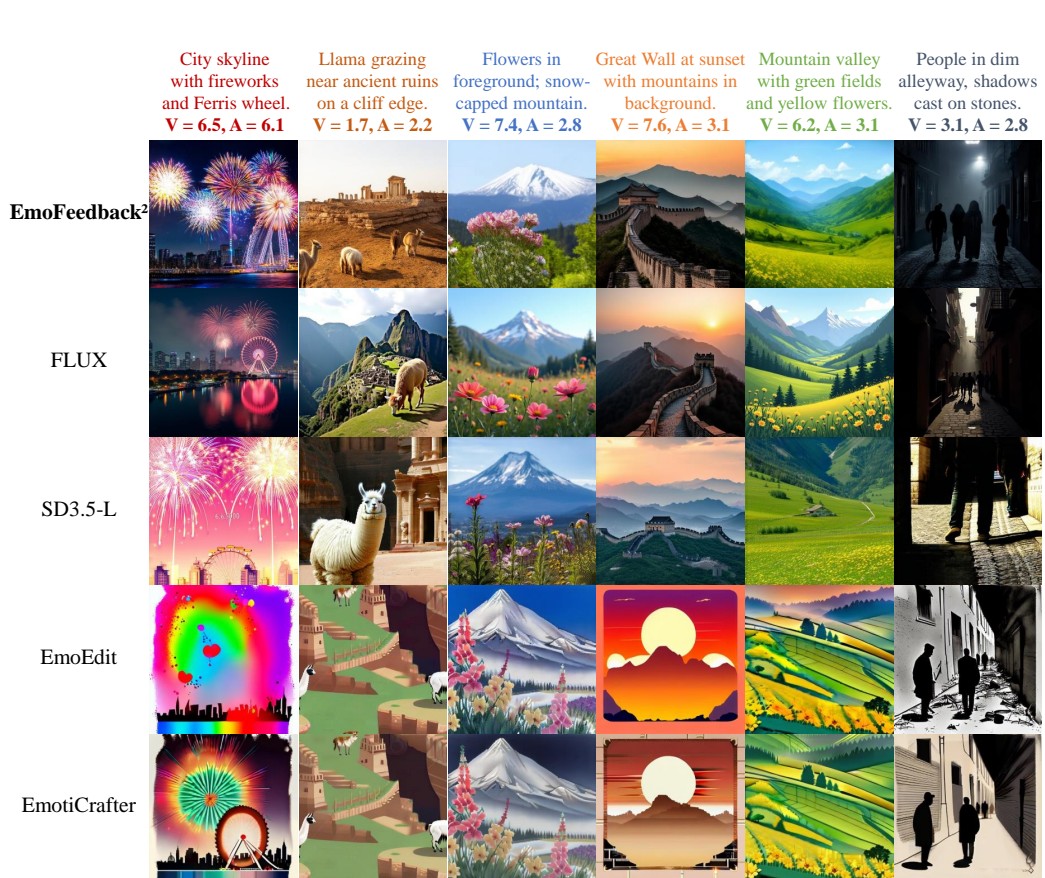

Figure 10: Additional Qualitative results under specific emotional states.

Valence Increase

Arousal Increase

Figure 11: Additional Qualitative results under varying emotional states. The original neutral prompt is "Christmas scene with miniature houses, trees, and a horseman". As valence and arousal increase, the light in this picture turns bright. The light in the cabin makes the atmosphere warmer.

Valence Increase

Arousal Increase

Figure 12: Additional Qualitative results under varying emotional states. The original neutral prompt is "Person sitting on storefront ledge, looking up". As valence and arousal increase, the person's facial expression changes from grave to a laugh. The light in the picture becomes brighter. The city is more colorful.

Valence Increase

Arousal Increase

Figure 13: Additional Qualitative results under varying emotional states. The original neutral prompt is "Skull and skeleton wearing a blue hooded cape outside". As valence and arousal increase, the skull's facial expression changed from frightening to amused. The added moon increases the light in the picture.

