# OpenReview forum: "EmoFeedback²: Reinforcement of Continuous Emotional Image Generation via LVLM-based Reward and Textual Feedback"
_ICLR.cc/2026/Conference — ICLR 2026 Conference Withdrawn Submission_

### Official Review · Reviewer_pvHG · 2025-10-27

**Soundness:** 2
**Presentation:** 2
**Contribution:** 2
**Rating:** 4
**Confidence:** 4

**Summary:**

This paper proposes EmoFeedback², a novel generation-understanding-feedback reinforcement paradigm for Continuous Emotional Image Content Generation (C-EICG). This method aims to adaptively adjust prompts to control the emotional continuity of the generations. It employs a Large Vision-Language Model (LVLM) to serve as an emotion understanding model. This LVLM provide emotion-aware reward feedback to reinforcement fine-tune the generative model. During inference, it adopts a self-promotion textual feedback framework to iteratively analyze generated images and refine the user prompt during inference. The results on a custom EmoSet-118K based dataset show that the proposed method outperforms other techniques.

**Strengths:**

### 1. Novel generation-understanding-feedback reinforcement paradigm

The whole paradigm is novel. It trains the LVLM as a reward model for emotion understanding and uses this feedback to guide the image generation. Furthermore, it can adaptively refine the prompt to enhance the emotional control.

### 2. Comprehensive experimental validation
The method achieves superior quantitative performance across five metrics, including V-Error and A-Error,  CLIP-Score and CLIP-IQA, against many modern baselines (EmotiCrafter, EmoEdit, FLUX, SD3.5-L).

**Weaknesses:**

### 1.The whole pipeline is over complex and heavy
The multi-stage process (diffusion -> RL-train LVLM to produce reward -> GRPO optimization for generation model with LVLM evaluation -> iterative textual feedback with LVLM) is extremely resource-intensive. Even at test time, the iterative self-promotion textual feedback requires multiple LVLM calls per image. Compared with using only the generation model to produce multiple outputs and get the best results, it seems a little unpractical.

### 2. Limited applicability
The model is trained on a custom dataset derived from EmoSet-118K with valence and arousal annotations, which might not generalize well to natural emotional cues. It is also unclear whether the model would perform well on out-of-distribution (OOD) scenarios.

**Questions:**

1. Sec4.1, which MLLM do you use for generating emotional prompts?
2. Line228 and Line194, you use epsilon for both reward threshold and clipping value of GRPO.
3. What is the batch size during RL training, and the number of epochs?

---

> ### Author Response · Authors · 2025-11-23
> **Response to the Reviewer pvHG (1/3)**
>
> We truly appreciate the time and effort Reviewer pvHG has devoted to reviewing our paper. The comments are very insightful and have been invaluable in helping us promote the quality of our work. We now address your comments in detail as follows:
>
> **W1: The whole pipeline is over complex and heavy**
>
> **A1**: We apologize for the confusion. The goal of our method is to address current key limitations, *including the lack of emotional feedback and insufficient adaptability*, and it remains **neither complex nor heavy** in practical use.
>
> We agree that the overall training pipeline contains multiple stages. However, we would like to clarify that most of the complexity lies in the training phase, whereas the computational cost remains fully manageable and the inference‑time latency is comparable to existing baselines. Additionally, *we would like to emphasize that the added components in our framework are not arbitrary architectural complexity but are necessary to address the limitations of prior work.*  Our LVLM-based generation–understanding–feedback paradigm provides a unified framework, which enables our model to achieve superior continuous emotion control and high-quality image generation. Below, we address the reviewer’s concerns from two perspectives :
>
> - **Manageable resource overhead in multi‑stage training**
>
>     Although our training pipeline contains several stages, the computational overhead remains modest. In this work, we fine‑tune an LVLM rather than full retraining, making the training process much more resource-efficient, taking around 20 hours on an 8‑GPU H20 machine. After this stage, the LVLM is frozen and serves purely as a reward model for fine‑tuning the diffusion generator. Notably, since we do not adopt iterative textual feedback during training, the GRPO optimization cost is similar to that of most reward‑model‑based reinforcement fine‑tuning methods, without additional iterative loops. Therefore, the overall training resource cost is not excessive.
>
>     More importantly, as shown in **Table 1** and **Figure 5**, leveraging a trained emotion‑understanding LVLM to provide emotional feedback substantially enhances emotional fidelity and generation quality.
>
> - **Comparable and acceptable test‑time latency for real‑world use**
>
>     Our self-promotion textual feedback framework provides configurable iterative refinement at test time. Users can choose to perform a single generation round or request additional refinement. In each round, the model produces a batch of images, allowing the user to either pick a satisfactory result immediately or wait for the next refinement step, depending on their preference for emotional fidelity. Since our model is fine-tuned with emotion-aware reward feedback, it's capable of generating competitive emotional images even in a single round.  For example, on a single H20 GPU, generating 8 images per round takes 11s, while the LVLM-based emotion evaluation and prompt refinement take an additional 9s. We compare the average inference latency per sample between our model and the baselines, and the results are reported in **Table R1** (**Table 7** in **Appendix C**). As shown, our inference latency is comparable to that of current state-of-the-art models. This efficiency is largely because our base generator, SD3.5‑M, contains only 2.5B parameters, making it much lighter than other generative models.
>
>     *Furthermore, without self-promotion textual feedback, using only the generation model to produce multiple outputs and get the best results, users may need to manually try multiple prompts, potentially consuming more resources overall.* Additionally, such manual selection introduces user-dependent variability and cannot guarantee consistent generation quality. Notably, this controlled increase in inference compute follows the widely adopted test time scaling strategies used in advanced LVLM systems such as OpenAI o1 [1] and DeepSeek R1 [2], where extended inference compute leads to boosted output quality. Our self-promotion textual feedback aligns with this trend and thereby achieves further performance gains with controlled extra inference cost, as demonstrated by the ablation experiments shown in **Figure 5**.
>
> **Table R1: Comparison of inference latency per sample and base generator model size.**
> | Metric | EmotiCrafter | SD3.5-L | FLUX | EmoFeedback² |
> | :--- | :---: | :---: | :---: | :---: |
> | Inference Latency (s) | 1.5 | 4.0 | 17.0 | **6.4** |
> | Model Size (B) | 3.5 | 8.0 | 12.0 | **2.5** |
>
> **[1] Jaech A, Kalai A, Lerer A, et al. Openai o1 system card[J]. arXiv preprint arXiv:2412.16720, 2024.**
>
> **[2] Guo D, Yang D, Zhang H, et al. Deepseek-r1 incentivizes reasoning in llms through reinforcement learning[J]. Nature, 2025, 645(8081): 633-638.**

---

> ### Author Response · Authors · 2025-11-23
> **Response to the Reviewer pvHG (2/3)**
>
> **W2: Limited applicability**
>
> **A2**:  To demonstrate our applicability, we clarify the **relationship** between valence–arousal (V‑A) annotations and natural emotional cues, and we further validate the generalization ability of our model under **cross‑dataset OOD scenarios**. Our detailed response is as follows:
>
> - **Relationship between V-A annotations and natural emotional cues**
>
>    The custom dataset is constructed to enrich natural emotional cues rather than replace them. There exists an intrinsic mapping between V-A annotations and natural emotional cues, and both are jointly utilized in our model training to support fine-grained emotion control. Specifically, the natural emotional cues consist of emotional categories and emotional descriptions, whereas the original EmoSet‑118K dataset provides only the emotional categories. To enhance these cues, we utilize an MLLM to generate descriptive emotional prompts. Moreover, our objective is continuous emotional image generation. However, discrete emotional cues alone are insufficient to represent the full spectrum of emotion values required for continuous control. Therefore, to construct our dataset, we map the natural emotional cues to continuous V-A scores using the emotion lexicon proposed in [1]. Furthermore, to ensure our model comprehends natural emotional cues, we incorporated both emotion categories and descriptions into the training process. For the emotion understanding model, our multi-task reinforcement fine-tuning strategy enables **simultaneous** V-A score regression and emotion category classification, thereby enhancing evaluation performance. For the generative model, we provide **both** emotional prompts and V-A scores to facilitate the learning of alignment between emotional cues and image content. Collectively, these strategies allow the model to perform continuous control based on V-A scores while maintaining robust generalization to natural emotional cues.
>
> - **Generalization ability under cross‑dataset OOD conditions**
>
>     We validated the model's generalization ability in a cross-dataset OOD scenario using the **EMOTIC dataset**. While EMOTIC provides continuous V-A annotations, it fundamentally differs from our custom dataset: EMOTIC’s annotations describe the emotions of people in the image (e.g., facial expressions, body posture), whereas our dataset focuses on the global emotional expression of the overall content and scene. Consequently, the emotion distribution in EMOTIC represents an unseen domain for our model. The quantitative cross-dataset results comparing our model against baselines are provided in the Appendix. As shown in **Table R2** (**Table 2** in the article), our method achieves SOTA performance in V-Error, A-Error, and CLIP-IQA, along with a competitive CLIP-Score. These results demonstrate that our method maintains superior performance in terms of both emotional fidelity and image quality even in OOD scenarios, highlighting a significant generalization advantage over current SOTA methods.
>
> **Table R2: Performance comparison across various methods on the EMOTIC dataset.**
> | Method | V-Error ↓ | A-Error ↓ | CLIP-Score ↑ | CLIP-IQA ↑ | Aes-Score ↑ |
> | :--- | :---: | :---: | :---: | :---: | :---: |
> | EmotiCrafter | 1.253 | *1.288* | 27.072 | 0.909 | 5.430 |
> | SD3.5-L | 1.131 | 1.300 | *27.704* | *0.930* | 5.302 |
> | FLUX | *1.047* | 1.395 | **27.877** | *0.930* | **5.597** |
> | EmoFeedback² | **0.849** | **0.669** | 27.410 | **0.938** | *5.480* |
>
> **[1] Warriner A B, Kuperman V, Brysbaert M. Norms of valence, arousal, and dominance for 13,915 English lemmas[J]. Behavior research methods, 2013, 45(4): 1191-1207.**

---

> ### Author Response · Authors · 2025-11-23
> **Response to the Reviewer pvHG (3/3)**
>
> We truly appreciate the time and effort Reviewer pvHG has devoted to reviewing our paper. The comments are very insightful and have been invaluable in helping us promote the quality of our work. We now address your question in detail as follows:
>
> **Q1: Sec4.1, which MLLM do you use for generating emotional prompts?**
>
> **A1:** We use **Qwen-2.5-VL-7B-Instruct** for generating emotional prompts.
>
> **Q2: Line228 and Line194, you use epsilon for both reward threshold and clipping value of GRPO.**
>
> **A2:** We apologize for the confusion caused by the typos in the article. In the revision, we have modified the reward threshold as $\tau$ for distinction, and we have also updated **Figure 2** to ensure consistent notations in the paper.
>
> **Q3. What is the batch size during RL training, and the number of epochs?**
>
> **A3:** During emotion understanding RL training, **our batch size is set to 16, and we train 5 epochs for convergence.** During image generation RL training, **we set the training process to 1000 steps, and the batch size for every step is set to 16.** We have clarified it in the revised version.

---

> ### Author Response · Authors · 2025-11-27
> **Reminder for Reviewer pvHG**
>
> Dear Reviewer pvHG,
>
> We hope that our responses could adequately address your concerns. As the discussion phase deadline approaches, we warmly welcome further discussion regarding any additional concerns that you may have, and we sincerely hope you can reconsider the rating accordingly.
>
> Thank you for the time and appreciation that you have dedicated to our work.
>
> Best regards,
>
> Authors of submission 7588

---

### Official Review · Reviewer_Q4TF · 2025-10-31

**Soundness:** 3
**Presentation:** 3
**Contribution:** 3
**Rating:** 6
**Confidence:** 3

**Summary:**

This paper introduces a new framework for continuous emotional image generation (C-EICG). It tackles the limitations of existing methods, namely their lack of feedback on the generated image's actual emotion and their poor adaptability to user prompts. The core innovation is using a Large Vision-Language Model (LVLM) to provide two types of feedback:
In training, an LVLM evaluates the emotional (Valence-Arousal) values of generated images, providing a reward signal to fine-tune the generative model using reinforcement learning.
In Inference, the LVLM iteratively analyzes images and provides textual suggestions to refine the prompt, improving emotional fidelity.

**Strengths:**

1. The core generation-understanding-feedback paradigm is a significant contribution. Unlike prior methods that use emotion as a one-way condition, this work creates a closed-loop system. Using an LVLM as both a reward model (for training) and a text-based optimizer (for inference) is a novel and powerful approach.

2. The paper's claims are supported by a set of experiments. Figure 4 and the appendix figures (e.g., 9-11) compellingly illustrate the model's ability to generate smooth and coherent transitions as V-A values change, a key goal of C-EICG that general T2I models (FLUX, SD3.5-L) fail to achieve.

3. The authors effectively demonstrate the necessity of both feedback mechanisms. Figure 5 provides a clear qualitative ablation of RF and TF , and Tables 3 & 4 validate the design choices for the emotion understanding model itself (e.g., model size, reward function, and multi-task training).

**Weaknesses:**

1. The primary weakness stems from the dataset construction. Both the textual prompts and the crucial V-A labels are synthetic. Prompts are MLLM-generated , and V-A values are sampled from Gaussian distributions derived from discrete emotion categories, not obtained from human annotators. This raises concerns about whether the model is truly aligned with human emotional perception or just with the biases of the synthetic data pipeline.

2. High Inference Cost: The Self-Promotion Textual Feedback framework, while effective, appears to be computationally expensive. The appendix states it uses 3 iterations, generating 8 images per iteration. This iterative process, which requires multiple calls to a 7B LVLM for analysis and prompt optimization , would result in significant latency, making real-time applications difficult.

**Questions:**

Given that the reward model was trained only on synthetically sampled V-A values, how do the authors expect it to perform on a dataset with genuine human V-A annotations? Is there a risk of domain mismatch, where the model has optimized for the dataset's statistical quirks rather than true emotional representation?

---

> ### Author Response · Authors · 2025-11-23
> **Response to the Reviewer Q4TF(1/2)**
>
> We appreciate the comments of the reviewer Q4TF, which are very helpful in enhancing the quality of the paper. We now list our detailed point-by-point responses to your comments as follows:
>
> **W1 (Q1): about aligned with human emotional perception**
>
> **A1:** **Our dataset construction is aligned with human emotional perception**. In addition, we conducted experiments on our in-house human-annotated dataset. As shown in **Figure 6**, the distribution of the sampled V–A values closely matches that of the in-house human annotations. **Table R1 (Table 6 in the article)** further shows that the model achieves comparable performance on both datasets, providing additional evidence that the model can align with true human emotions without introducing the risk of domain mismatch.
>
> - **Dataset construction aligns with human verification**
>
>     Although our V–A labels are sampled from a lexicon, the lexicon itself is grounded in large-scale human emotion studies involving 1,827 annotators [1]. Consequently, the sampled V–A values effectively reflect genuine human emotional judgments. In addition, we already conducted manual verification and correction for all generated prompts during dataset construction. This process included rewriting prompts whose emotional content was inconsistent with the corresponding image, translating non-English prompts into English, simplifying overly neutral prompts, and removing neutral or contradictory words from emotional prompts. Together, these procedures ensure that our dataset construction remains closely aligned with human emotional perception.
>
>
> - **Domain consistency between in-house human annotations and sampled V-A values**
>
>     For V-A annotations, we engaged eight expert annotators to independently assign the V-A scores for all 1,000 test images in our custom dataset, using a 1–9 rating scale for both dimensions. To validate perceptual domain consistency, we compare the sorted distributions of the lexicon-sampled V-A values with the averaged human annotations. As shown in **Figure 6**, the two curves exhibit highly similar shapes and closely aligned monotonic trends for both dimensions. This distributional alignment strongly indicates that our lexicon-based synthetic labels preserve the global structure of human emotional perception, rather than matching merely on isolated values, demonstrating the reliability of our sampling scheme.
>
> - **Performance of the reward model on human-annotated data**
>
>     We computed the mean absolute error (MAE) between the reward model’s predicted V–A scores and the in-house human-annotated labels (reported as V-Error and A-Error). As shown in **Table R3 (Table 6 in the article)**, the reward model achieves MAE values comparable to those obtained with lexicon-based labels, demonstrating that our model is truly aligned with human emotional perception. As the lexicon encodes emotions aggregated over a large and diverse human population, the reward model learns a generalized emotional representation that reflects population-level tendencies rather than overfitting to statistical quirks of the synthetic samples.
>
> **Table R1: Comparison between Lexicon-based Annotation and Human Annotation**
>
> | Annotator | Lexicon-based Annotation | Human Annotation |
> | :--- | :---: | :---: |
> | V-Error | 0.521 | 0.781 |
> | A-Error | 0.710 | 1.310 |
>
> **[1] Warriner A B, Kuperman V, Brysbaert M. Norms of valence, arousal, and dominance for 13,915 English lemmas[J]. Behavior research methods, 2013, 45(4): 1191-1207.**

---

> ### Author Response · Authors · 2025-11-23
> **Response to the Reviewer Q4TF(2/2)**
>
> **W2:High Inference Cost**
>
> **A2:** Although our framework incorporates iterative refinement, we would like to clarify that **the iterative process is deliberately designed to address limitations in prior work**, and **its inference latency remains comparable with other models and acceptable for real-time use**. We elaborate on these points below.
>
> - **Necessary for overcoming current limitations**
>
>     It is important to emphasize that the iterative textual feedback mechanism addresses a key limitation of existing methods: they are unable to adaptively refine prompts based on the actual image content. Without our refinement process, users typically need to try several prompts manually, resulting in higher total latency than our guided iterative strategy. As demonstrated in **Figure 5**, our refinement process enhances fine-grained emotional details and yields more faithful emotional expression. Notably, this controlled increase in inference cost follows the widely adopted test time scaling strategies used in advanced LVLM systems such as OpenAI o1 [1] and DeepSeek R1 [2], where extended inference cost leads to boosted output quality.
>
> - **Average latency is comparable to other generative models**
>
>     To further demonstrate the feasibility of our approach, we quantitatively compare our average inference latency per sample against baseline models in **Table R2 (Table 7 in the Appendix C)**. As shown, despite the use of an LVLM, our latency remains comparable to state-of-the-art text-to-image generators. A key factor enabling this efficiency is our lightweight base generator SD3.5-M (2.5B parameters), which is substantially smaller than other diffusion backbones and makes our system suitable for real-world deployment.
>
> - **Flexible inference cost is practically acceptable to users**
>
>     The iterative procedure in our self-promotion textual feedback framework is not mandatory. In each round, the model produces a batch of images, and users may immediately select a satisfactory result or wait for further refinement, depending on their quality requirements. Since our model is fine-tuned with emotion-aware reward feedback, it can already produce competitive emotional images in a single iteration. In our standard experimental setting (3 iterations and 8 images on a single H20 GPU), the generation process requires 11s, while the LVLM evaluates the eight images in parallel and refines the prompt in 9s. Such flexible and user-controllable latency is generally acceptable in practical scenarios.
>
> **Table R2: Comparison of inference latency per sample and base generator model size.**
>
> | Metric | EmotiCrafter | SD3.5-L | FLUX | EmoFeedback² |
> | :--- | :---: | :---: | :---: | :---: |
> | Inference Latency (s) | 1.5| 4.0 | 17.0 | **6.4** |
> | Model Size (B) | 3.5 | 8.0 | 12.0 | **2.5** |
>
> **[1] Jaech A, Kalai A, Lerer A, et al. Openai o1 system card[J]. arXiv preprint arXiv:2412.16720, 2024.**
>
> **[2] Guo D, Yang D, Zhang H, et al. Deepseek-r1 incentivizes reasoning in llms through reinforcement learning[J]. Nature, 2025, 645(8081): 633-638.**

---

> ### Author Response · Authors · 2025-11-27
> **Reminder for Reviewer Q4TF**
>
> Dear Reviewer Q4TF,
>
> We hope that our responses could adequately address your concerns. As the discussion phase deadline approaches, we warmly welcome further discussion regarding any additional concerns that you may have, and we sincerely hope you can reconsider the rating accordingly.
>
> Thank you for the time and appreciation that you have dedicated to our work.
>
> Best regards,
>
> Authors of submission 7588

---

### Official Review · Reviewer_Zyi6 · 2025-10-31

**Soundness:** 2
**Presentation:** 3
**Contribution:** 2
**Rating:** 4
**Confidence:** 4

**Summary:**

This paper proposes a reinforcement-based framework for continuous emotional image generation. The method integrates a Large Vision–Language Model (LVLM) as both a reward model and a textual feedback generator, enabling a closed loop of generation, understanding, and feedback. Through emotion-aware reward optimization and iterative prompt refinement, the approach enhances emotional continuity and fidelity.

**Strengths:**

1. The generation–understanding–feedback paradigm effectively unites emotional reasoning with reinforcement learning in diffusion models.

2. The textual feedback loop provides an intuitive and interpretable way to refine emotional prompts beyond fixed embeddings.

3. The method achieves the best V-Error, A-Error, CLIP-Score, and user preference rate, validating both emotional fidelity and visual quality.

**Weaknesses:**

1. The dataset construction strategy closely follows EmotiCrafter (Dang et al., 2025), which already employed an MLLM to generate neutral and emotional prompts as well as Valence–Arousal annotations, resulting in limited methodological novelty.

2. The ground-truth Valence–Arousal annotations are sampled from Gaussian distributions per emotion class, which can introduce label noise and semantic misalignment between images and emotional values.

3. The constructed dataset depends entirely on automatically generated prompts and lexicon-based statistical sampling without any human verification, raising concerns about annotation reliability and perceptual validity.

4. The proposed framework heavily relies on the large vision–language model (LVLM) for both reward computation and textual feedback generation, making the entire pipeline sensitive to model bias and reasoning instability.

5. The self-promotion textual feedback requires iterative LVLM reasoning over multiple generated samples, which substantially increases inference cost, yet the paper does not provide any discussion or analysis of computational efficiency or scalability.

**Questions:**

1. Given that the ground-truth Valence–Arousal annotations are sampled from Gaussian distributions per emotion class, how do the authors ensure that such synthetic labels accurately reflect the emotional content of the images and do not introduce semantic noise?

2. As the dataset relies entirely on automatically generated prompts and lexicon-based annotations without any human verification, have the authors conducted any manual inspection or validation to assess label reliability and perceptual consistency?

3. The proposed framework depends heavily on the LVLM for both reward computation and textual feedback generation. How robust is the system to the biases or reasoning instability of the LVLM, and could similar performance be achieved with a smaller or less powerful model?

4. The iterative textual feedback mechanism requires multi-image evaluation and textual generation at each step. Could the authors provide quantitative analysis on inference time, computational cost, or scalability to demonstrate its practical feasibility?

5. In Figure 3, the results produced by EmoEdit and EmotiCrafter appear highly similar. Could the authors clarify whether this similarity arises from shared input prompts, overlapping training data, or something else?

---

> ### Author Response · Authors · 2025-11-23
> **Response to Reviewer Zyi6(1/6)**
>
> We thank the reviewer Zyi6 for the thorough and insightful evaluation of our paper, which has significantly improved our manuscript. In response to the reviewer's comments, we have made substantial revisions and improvements, as detailed below.
>
> **W1: about dataset construction strategy close to EmotiCrafter**
>
> **A1:** We apologize for the confusion. Our methodologies diverge fundamentally in both **objective** and **implementation**.
>
> - **Objective:**
> While both our work and EmotiCrafter [1] leverage MLLM to generate neutral and emotional prompts, EmotiCrafter essentially treats emotional prompts as fixed ground-truth targets for V-A mapping, whereas our method utilizes emotional prompts directly as inputs to the generative model.
>
> - **Implementation:**
> EmotiCrafter employs an emotion-embedding network to pair V-A values with neutral textual features, aligning them with emotional textual features via loss constraints. However, binding emotions to static textual representations prevents them from adaptively refining emotional prompts based on image content, limiting generation diversity and emotional fidelity.
> In contrast, our model learns a dynamic mapping between emotional semantics and visual contents. It trains the generative model directly on triplet inputs (V-A values, neutral prompts, and emotional prompts), enabling the model to learn dynamic relationships between emotional semantics and visual content. During test time, we only input V-A values and neutral prompts. Additionally, to flexibly supply emotional prompts, we design a novel self-promotion textual feedback framework for multi-round generation, which iteratively analyzes emotional content and adaptively refines emotional prompts for the next-round generation. This closed-loop optimization enriches fine-grained details to enhance both adaptability and emotional fidelity, which EmotiCrafter cannot support.
>
> Thus, although both works rely on MLLMs to initialize text descriptions, our framework introduces a fundamentally new paradigm. In particular, the self-promotion textual feedback framework enables adaptive prompt refinement driven by iterative LVLM-based analysis, which advances beyond current emotion-text alignment approaches. This design introduces methodological novelty in the **generation–understanding–feedback loop**, extending far beyond dataset construction alone. We have revised the manuscript to explicitly state this distinction and to better highlight our contributions.
>
> **[1]. Dang S, He Y, Ling L, et al. Emoticrafter: Text-to-emotional-image generation based on valence-arousal model[C]//Proceedings of the IEEE/CVF International Conference on Computer Vision. 2025: 15218-15228.**

---

> ### Author Response · Authors · 2025-11-23
> **Response to Reviewer Zyi6(2/6)**
>
> **W2 (Q1): about label noise and semantic misalignment**
>
> **A1:** We apologize for the confusion. We explain the rationality of the V–A annotations from three perspectives: **the sampling aligns with the real scenarios**, **the implementation strategy mitigates label noise**, and **the experimental results demonstrate the effectiveness**.
>
> - **The sampling aligns with the real scenarios**
>
>     The emotion category labels in the dataset are derived from the aggregated distribution formed by 1,827 annotators [1], where each point in the distribution represents a single annotated sample. The use of Valence–Arousal annotations sampled from Gaussian distributions is intended to simulate real-world multimodal perceptual variability. Therefore, our sampling strategy is both appropriate and well-motivated, introducing no additional label noise.
>
> - **The implementation strategy mitigates label noise**
>
>     To mitigate potential label noise and semantic misalignment introduced by sampling, we adopt a multi-task training strategy for the emotion understanding model. This approach jointly trains the model through a regression task for V-A scores and a classification task for emotion categories. The classification task ensures precise alignment between image content and emotion category labels, while Gaussian sampling introduces minor perturbations to the established alignment. This design preserves the inherent ambiguity of affective judgments without inducing severe noise or semantic drift.
>
> - **The experimental results demonstrate the effectiveness**
>
>     In our ablation study **Table R1 (Table 5 in the article)**, we compare the performance of emotion understanding models under regression-only, classification-only, and joint training strategies. Results demonstrate that the multi-task model achieves significantly lower V-Error and A-Error than single-task models. This confirms that the combination of discrete emotion labels and Gaussian-sampled continuous V-A targets can accurately reflect the emotional content of images with minimal semantic noise, while mitigating overfitting to discrete points, synergistically enhancing model performance.
>
> **Table R1: Ablation Study between multi-task and single-task**
>
> | Task | Jointly Training | Regression Only | Classification Only |
> | :--- | :---: | :---: | :---: |
> | V-Error | **0.521** | 0.579 | 1.445 |
> | A-Error | **0.710** | 0.812 | 2.073 |
>
> **[1] Warriner A B, Kuperman V, Brysbaert M. Norms of valence, arousal, and dominance for 13,915 English lemmas[J]. Behavior research methods, 2013, 45(4): 1191-1207.**

---

> ### Author Response · Authors · 2025-11-23
> **Response to Reviewer Zyi6(3/6)**
>
> **W3 (Q2): about annotation reliability and perceptual validity**
>
> **A3:** We apologize for the confusion. In fact, our method incorporates **human verification**, and we explain this from the following **three** perspectives:
>
> - **Dataset construction aligns with human verification**
>
>     Although our V–A labels are sampled from a lexicon, the lexicon itself is derived from large-scale human emotion studies involving 1,827 annotators [1]. Thus, the sampled V–A values can reflect true human emotional judgments. For generated prompts, we already performed manual verification and correction for them during dataset construction. This included human rewriting prompts inconsistent with the image’s emotional content, converting non-English prompts to English, simplifying neutral prompts, and removing neutral or contradictory words from emotional prompts.
>
> - **Distributional consistency between in-house human annotations and lexicon-based labels**
>
>     We conducted a series of human verification studies involving expert annotators. Our findings demonstrate that the lexicon-based labels are largely consistent with the human perception of our in-house annotations.
>     For Valence (V) and Arousal (A) annotations, we asked eight expert annotators to independently label the V-A scores for all 1,000 test images in our custom dataset, using a 1–9 rating scale for both dimensions. To validate perceptual consistency, we compared the sorted distributions of the lexicon-based V–A annotations and the averaged human annotations on the 1,000 test images. As shown in **Figure 6**, the two distributions exhibit highly similar shapes and closely aligned monotonic trends for both dimensions. This distributional alignment strongly indicates that our lexicon-based annotations preserve the global structure of human emotional perception, rather than matching only on isolated values, demonstrating the reliability of our labeling scheme.
>
> - **Comparable performance on in-house human-annotated labels**
>
>     We further evaluated label reliability and perceptual consistency by comparing the emotion understanding model’s performance on in-house human-annotated labels and lexicon-based labels. As shown in **Table R2 (Table 6 in the article)**, their V-Error and A-Error are comparable, demonstrating that our model is truly aligned with human emotional perception.
>
> **Table R2: Comparison between Lexicon-based Annotation and Human Annotation**
>
> | Annotator | Lexicon-based Annotation | Human Annotation |
> | :--- | :---: | :---: |
> | V-Error | 0.521 | 0.781 |
> | A-Error | 0.710 | 1.310 |
>
> **[1] Warriner A B, Kuperman V, Brysbaert M. Norms of valence, arousal, and dominance for 13,915 English lemmas[J]. Behavior research methods, 2013, 45(4): 1191-1207.**

---

> ### Author Response · Authors · 2025-11-23
> **Response to Reviewer Zyi6(4/6)**
>
> **W4: (Q3): about model bias and reasoning instability**
>
> **A4:** We elaborate on how our method addresses LVLM bias and reasoning instability, and whether similar performance can be achieved with smaller models.
>
> - **How robust is the system to the biases or reasoning instability of the LVLM**
>
>     We fully recognize that a generic LVLM may exhibit bias or unstable reasoning in certain tasks. To address this, we do not directly use an off-the-shelf LVLM for reward computation or feedback generation. Instead, we build a task-specialized Emotion Understanding Model (EUM). Specifically, we adopt Qwen2.5-VL-7B-Instruct as the backbone and apply Group Relative Policy Optimization (GRPO) with a multi-task reward design, including V–A score regression, and emotion classification. Through this emotion-aligned reinforcement fine-tuning, the LVLM’s behavior is guided and constrained to focus solely on accurate emotional assessment, rather than relying on its potentially unstable or biased reasoning. The experimental results in **Table R3** demonstrate that our task-specialized EUM achieves significantly better performance than a generic LVLM. Additionally, ablation studies show performance degradation without LVLM components (**Figure 5**), indicating the robust contributions of LVLM.
>
> - **Performance comparison with a smaller LVLM**
>
>     To examine the impact of model size, we conducted a quantitative comparison in Section 4.4.1. We trained a smaller EUM based on Qwen2.5-VL-3B-Instruct (Qwen-3B-S) and compared it with our 7B model. As shown in **Table R4 (Table 4** in the article), performance drops markedly with the smaller model: V-Error from 0.521 (Ours, 7B) to 0.628 (Qwen-3B-S), A-Error from 0.710 (Ours, 7B) to 1.217 (Qwen-3B-S). These results clearly indicate that while a smaller LVLM can serve as the EUM, the loss in accuracy significantly degrades both reward feedback and textual refinement quality, ultimately harming the emotional fidelity of the generated images. Therefore, the 7B LVLM represents a balanced choice, offering strong performance while keeping computational demands practical.
>
> **Table R3: Performance comparison between generic LVLM and our EUM.**
> | Method | V-Error ↓ | A-Error ↓ | CLIP-Score ↑ | CLIP-IQA ↑ | Aes-Score ↑ |
> | :--- | :---: | :---: | :---: | :---: | :---: |
> | General | 0.850 | 0.842 | 26.776 | 0.821 | 5.399 |
> | Emotion | **0.521** | **0.710** | **26.889** | **0.880** | **5.442** |
>
>
> **Table R4: Ablation Study between different reward functions and model size.**
>
> | Model | Ours | Qwen-3B-S | Qwen-7B-C |
> | :--- | :---: | :---: | :---: |
> | V-Error | **0.521** | 0.628 | 0.819 |
> | A-Error | **0.710** | 1.217 | 0.896 |

---

> ### Author Response · Authors · 2025-11-23
> **Response to Reviewer Zyi6(5/6)**
>
> **W5 (Q4): about computational efficiency**
>
> **A5:** Our self-promotion textual feedback framework is designed to address a **key limitation** of existing methods that they cannot adaptively refine low-quality prompts during inference. Moreover, its **computational efficiency** is comparable to that of other models and is acceptable for practical use. The relevant analysis is presented below:
>
> - **Necessity of our framework**
>
>     Existing methods focus on single-round generation and cannot adaptively refine low-quality prompts during inference. Users often need to manually test multiple prompts, which can ultimately consume more computational resources. To address this limitation, our self-promotion textual feedback framework iteratively analyzes the emotional content of generated images and produces refinement suggestions for the next-round prompt, improving emotional fidelity with fine-grained content.
>
> - **Average latency is comparable to other generative models**
>
>     To provide a quantitative comparison of practical efficiency, we report the average inference latency per sample and the base generator model size for our method and baseline models. As shown in **Table R5 (Table 7 in Appendix C)**, despite employing an LVLM, our inference time is comparable to state-of-the-art generative frameworks. This efficiency is largely due to our lightweight base generator, SD3.5-M (2.5B parameters), which is substantially smaller than many other diffusion backbones and makes our system suitable for real-world deployment.
>
> - **Flexible inference cost is practically acceptable to users**
>
>     Our framework allows flexibility in both the number of images per iteration and the number of refinement iterations. In each round, the model produces a batch of images, and users may immediately select a satisfactory result or wait for further refinement, depending on their quality requirements. Since our model is fine-tuned with emotion-aware reward feedback, it can already produce competitive emotional images in a single iteration. In our standard setting of three iterations with eight images, the step-wise time cost measured on a single H20 GPU shows that the image generation stage takes 11s, while the LVLM-based emotion analysis and prompt refinement require 9s. Such flexible and user-controllable latency is generally acceptable in practical scenarios.
>
> **Table R5: Comparison of inference latency per sample and base generator model size.**
>
> | Metric | EmotiCrafter | SD3.5-L | FLUX | EmoFeedback² |
> | :--- | :---: | :---: | :---: | :---: |
> | Inference Latency (s) | 1.5 | 4.0 | 17.0 | **6.4** |
> | Model Size (B) | 3.5 | 8.0 | 12.0 | **2.5** |

---

> ### Author Response · Authors · 2025-11-23
> **Response to Reviewer Zyi6(6/6)**
>
> **Q5: about EmoEdit and EmotiCrafter**
>
> **A6:** The high similarity primarily arises from the fact that EmoEdit is an editing model, and the images it edits are generated by **the same SDXL generator** as EmotiCrafter.
>
> For a fair comparison, we provided both EmotiCrafter and EmoEdit with **the same text prompts** as those used for our method, and we strictly followed the official pipelines of both approaches. EmotiCrafter is a text-to-image generation model that adopts SDXL as its base generator and learns an emotion embedding network to inject Valence–Arousal (V–A) values. EmoEdit, on the other hand, is an image editing model that introduces an Emotion Adapter to modify emotional content based on an input image and a discrete emotion label. Since EmoEdit requires an image as input and our comparison setting does not supply one, we first generated images with **the same SDXL generator**, and then applied EmoEdit to modify their emotional content. Consequently, *both methods operate on SDXL-generated images, naturally leading to highly similar visual styles*.
>
> This similarity also reflects a limitation shared by the two methods: they rely solely on an additional emotion-mapping network, but lack mechanisms for providing feedback on the emotional content of the generated images. Consequently,  they tend to produce homogeneous outputs with limited diversity. In contrast, our framework incorporates both reward and textual feedback, enabling fine-grained emotional control and producing much more diverse emotional content, as illustrated in **Figure 3**.

---

> ### Author Response · Authors · 2025-11-27
> **Reminder for Reviewer Zyi6**
>
> Dear Reviewer Zyi6,
>
> We hope that our responses could adequately address your concerns. As the discussion phase deadline approaches, we warmly welcome further discussion regarding any additional concerns that you may have, and we sincerely hope you can reconsider the rating accordingly.
>
> Thank you for the time and appreciation that you have dedicated to our work.
>
> Best regards,
>
> Authors of submission 7588

---

### Official Review · Reviewer_jgo1 · 2025-11-01

**Soundness:** 3
**Presentation:** 3
**Contribution:** 3
**Rating:** 6
**Confidence:** 4

**Summary:**

This paper proposes a generation-understanding-feedback reinforcement paradigm for continuous emotional image generation (C-EICG). It leverages a fine-tuned Large Vision-Language Model (LVLM) to address key limitations of existing methods—lack of emotional feedback and insufficient adaptability of emotional prompts—by introducing an emotion-aware reward feedback strategy and a self-promotion textual feedback framework, aiming to enhance emotional continuity and fidelity while maintaining image quality.

**Strengths:**

1. **Novel Paradigm Design**: The proposed "generation-understanding-feedback" reinforcement framework fills a gap in C-EICG by integrating emotional feedback loops. Unlike existing methods that ignore post-generation emotional evaluation, it uses LVLM’s reasoning ability to close the loop between generation and optimization, bringing a new perspective to emotional image generation.
2. **Well-Integrated Multi-Module Collaboration**: The emotion understanding model (fine-tuned via GRPO), emotion-aware reward feedback, and self-promotion textual feedback are logically coordinated. Each module addresses a specific pain point, and their synergy ensures both emotional accuracy and content consistency, demonstrating a coherent design philosophy.
3. **Practical Training-Free Inference Optimization**: The self-promotion textual feedback framework enables adaptive prompt refinement during inference without retraining the generative model. This design enhances usability, as it can be easily integrated with existing diffusion models (e.g., Stable Diffusion 3.5-Medium) without heavy parameter tuning.
4. **Comprehensive Ablation Studies**: Ablation experiments on LVLM size, reward function design, and single/multi-task training effectively validate the necessity of key design choices. These studies clarify the contribution of each component, strengthening the credibility of the proposed method.
5. **Rich Evaluation Dimensions**: Beyond standard quantitative metrics (emotional accuracy, text-image alignment), the paper includes qualitative comparisons and user studies. This multi-faceted evaluation better reflects the method’s performance in real-world scenarios, aligning with the subjective nature of emotional perception.

**Weaknesses:**

1. **Limited Discussion on LVLM’s Emotional Evaluation Mechanism**: The paper does not deeply explain how the LVLM (Qwen2.5-VL-7B-Instruct) specifically interprets visual content to assess emotions. The lack of analysis on which visual cues (e.g., color, composition) the LVLM prioritizes makes it hard to understand the mechanistic advantage of using LVLM for emotional feedback.
2. **Insufficient Generalization Validation**: The experiments are primarily conducted on a custom dataset derived from EmoSet-118K. There is no validation on other public C-EICG datasets or cross-domain scenarios (e.g., different image styles, complex scenes), raising questions about the method’s generalizability.
3. **Vague Explanation of Reward Hacking Mitigation**: While the paper mentions using PickScore to avoid content distortion caused by overfitting to emotional cues, it does not detail how PickScore is integrated with the reward function or why it is more effective than other human-preference metrics. This makes the mitigation strategy less transparent.
4. **Lack of Comparison with LVLM-Based Alternatives**: With the rise of LVLM-driven generation optimization, the paper does not compare EmoFeedback2 with other LVLM-aided C-EICG methods (if any) or analyze its unique advantages over general LVLM-based feedback frameworks, weakening the demonstration of its competitiveness.

**Questions:**

Please refer to the detailed points I raised in the "Weakness" section and respond to each numbered item in your rebuttal with clarifications.

---

> ### Author Response · Authors · 2025-11-23
> **Response to Reviewer jgo1(1/4)**
>
> We would like to express our sincere gratitude to Reviewer jgo1’s professional review, which is instrumental in improving the clarity and quality of our paper. We now list our detailed point-by-point responses to your comments as follows:
>
> **W1: Limited Discussion on LVLM’s Emotional Evaluation Mechanism**
>
> **A1:** In the revised version, we have added a detailed discussion of LVLM’s emotional evaluation mechanism in the **Appendix D**.
>
> LVLMs benefit from large-scale multimodal pretraining on diverse image–text pairs that frequently contain affective descriptions, enabling them to learn rich associations between visual patterns and emotional semantics. This provides a strong foundation for emotion understanding. In this work, we mainly demonstrate the process and rationale of LVLM-based emotion assessment through explicit prompt guidance and chain-of-thought reasoning.
>
> - **Explicit prompt guidance**: Leveraging the instruction-following and multimodal alignment mechanisms of LVLMs to focus on key visual cues.
>
>     The LVLM is not treated as a complete “black box”. Instead, we actively guide its attention towards emotion-relevant cues through carefully designed prompts. As shown in **Appendix D (Table 8)**, the prompts used during training and evaluation explicitly instruct the model to focus on visual elements highly relevant to emotion, such as: *“Please consider the weather, light, background object, and facial expression in the decision.”* The multimodal alignment mechanism of LVLMs enables the model to prioritize these features in visual encoding, which are known to be important in human emotional perception. The explicit prompts can introduce inductive bias that significantly influences the distribution of attention weights in LVLMs [1]. Therefore, by leveraging these properties, our model improves both the controllability and interpretability of LVLM’s emotion assessment and ensures that the evaluation process is grounded in meaningful and human-aligned visual attributes.
>
> - **Chain-of-thought reasoning**: Utilizing the reasoning capabilities and hierarchical feature extraction of LVLM to enable the emotional attribution process to be interpretable.
>
>     We instruct the model to explicitly output its reasoning steps, making emotion judgment no longer implicitly encoded but expressed through an interpretable reasoning path. **Figure 2** provides a concrete example. Before outputting the final emotion scores, the model clearly states its reasoning: *(1) identifying key visual elements such as “castle,” “flowers,” and “bunny ears”; (2) interpreting visual attributes such as “bright” referring to color, “playful” referring to style, and “surrounded” referring to composition; (3) linking these elements to emotional implications, such as “amusement,” “positive,” “excitement”. Through hierarchical feature extraction, the LVLM simultaneously captures low-level visual cues and high-level element attributes and uses cross-modal associations to map visual features into an abstract emotional semantic space.* This leads to a coherent reasoning process that greatly enhances the reliability and interpretability of emotional inference.
>
> In summary, by combining explicit prompt guidance and chain-of-thought reasoning, our model leverages the multimodal alignment and hierarchical feature extraction strengths of LVLMs, enabling the model to selectively attend to emotionally relevant visual cues and produce more controllable, interpretable, and reliable emotional feedback. In the revised manuscript, we have expanded the discussion of these mechanisms more explicitly.
>
> **[1]. Wei J, Wang X, Schuurmans D, et al. Chain-of-thought prompting elicits reasoning in large language models[J]. Advances in neural information processing systems, 2022, 35: 24824-24837.**

---

> ### Author Response · Authors · 2025-11-23
> **Response to Reviewer jgo1(2/4)**
>
> **W2: Insufficient Generalization Validation**
>
> **A2:** Here, we first clarify **why** our initial experiments focused only on a custom dataset, and then we provide additional validation of the **cross‑dataset generalization** ability of our method.
> - In this work, we aim to generate images whose content continuously varies with emotional values. The construction of a custom dataset (**Section 4.1**) was designed primarily due to these task-specific requirements. **Specifically, the C‑EICG task requires triplets of text, image, and continuous Valence-Arousal (V-A) values.** However, existing public datasets such as EmoSet‑118K typically contain only images paired with discrete emotion categories, making them incapable of supporting continuous emotional supervision. While datasets like EMOTIC [1] and FindingEmo [2] provide continuous V-A labels, their annotations capture the emotional states of people in the image (e.g., facial expressions, body posture) rather than the emotional expression of the entire image in terms of content and scene. This differs from our task objective of generating content that continuously varies with emotions. Therefore, our primary experiments relied on a custom dataset aligned with the task requirements.
>
> - As suggested by the reviewer, we fully recognized the importance of evaluating its **cross‑dataset generalization**. Although EMOTIC focuses on human‑centered emotion labels, its continuous V‑A annotation system can still serve as an external reference for testing the model’s emotional understanding and generation ability on cross‑domain images. Thus, we conducted an additional evaluation on the EMOTIC test set, and the comparison results are now reported in **Table R1** (**Table 2** in the article).  Our method achieves the best V‑Error, A‑Error, CLIP‑IQA, and competitive CLIP‑Score, demonstrating strong generalization in both emotional fidelity and visual quality. Given the smaller capacity of our backbone model SD3.5‑M (2.5B) compared with that of FLUX (12B), our method obtains slightly lower Aes-Score than FLUX’s best results.
>
> **Table R1: Performance comparison across various methods on the EMOTIC dataset.**
> | Method | V-Error ↓ | A-Error ↓ | CLIP-Score ↑ | CLIP-IQA ↑ | Aes-Score ↑ |
> | :--- | :---: | :---: | :---: | :---: | :---: |
> | EmotiCrafter | 1.253 | *1.288* | 27.072 | 0.909 | 5.430 |
> | SD3.5-L | 1.131 | 1.300 | *27.704* | *0.930* | 5.302 |
> | FLUX | *1.047* | 1.395 | **27.877** | *0.930* | **5.597** |
> | EmoFeedback² | **0.849** | **0.669** | 27.410 | **0.938** | *5.480* |
>
> **[1]. Kosti R, Alvarez J M, Recasens A, et al. Context based emotion recognition using emotic dataset[J]. IEEE transactions on pattern analysis and machine intelligence, 2019, 42(11): 2755-2766.**
>
> **[2]. Mertens L, Yargholi E, Op de Beeck H, et al. Findingemo: An image dataset for emotion recognition in the wild[J]. Advances in Neural Information Processing Systems, 2024, 37: 4956-4996.**

---

> ### Author Response · Authors · 2025-11-23
> **Response to Reviewer jgo1(3/4)**
>
> **W3: Vague Explanation of Reward Hacking Mitigation**
>
> **A3:** We now elaborate on **how** PickScore is integrated and **why** it is more effective than other human-preference metrics, followed by an **ablation study** that demonstrates its impact on generation quality.
>
> - **How it is integrated**: PickScore is incorporated as an additional reward term in the GRPO-based model. reinforcement learning framework. The combined reward is defined as the weighted sum of the emotion fidelity reward (derived from our emotion-understanding LVLM) and the human preference reward (PickScore).
> - **Why PickScore**: In our context, reward hacking manifests when the model sacrifices image quality or semantic content to maximize the emotion reward (e.g., generating a solid red patch to represent "anger").  PickScore mitigates the reward hacking by providing a strong human-preference prior. Specifically, PickScore is a CLIP-based scoring model trained on a substantially larger and more diverse human preference dataset than alternative human-preference metrics such as ImageReward and HPS. As demonstrated by [1], PickScore achieves the highest correlation with human judgments in terms of both image-text alignment and aesthetic quality. Thus, the strong human-aligned evaluation ability enables effective penalization of content distortion.  If the model distorts the original prompt in pursuit of emotional expression (e.g., a "dog" no longer resembles a dog), PickScore assigns a significantly lower score, thereby offsetting the high Emotion Reward. This mechanism forces the model to strike a balance between emotional fidelity and semantic consistency/aesthetic quality.
>
> - To further validate the necessity of PickScore, we trained a variant of EmoFeedback² without the PickScore reward for **ablation study**. We compared sample images generated by both models at various training stages. As shown in **Figure 7 in Appendix E.1.1**, images generated by the model trained without PickScore gradually lose semantic features as training progresses. Conversely, the model trained with the multi-reward objective maintains semantic consistency and image quality.
>
> **[1] Kirstain Y, Polyak A, Singer U, et al. Pick-a-pic: An open dataset of user preferences for text-to-image generation[J]. Advances in neural information processing systems, 2023, 36: 36652-36663.**

---

> ### Author Response · Authors · 2025-11-23
> **Response to Reviewer jgo1(4/4)**
>
> **W4: Lack of Comparison with LVLM-Based Alternatives**
>
> **A4:** To the best of our knowledge, there have been **no** other LVLM‑aided C-EICG methods proposed to date, and our work is among the **first** explorations in this area. Therefore, we analyze the advantages of our method over general LVLM-based frameworks in **two** key aspects, and conduct **comparative experiments** for validation:
>
> - **Emotion‑specialized LVLM**
>
>     Most existing feedback frameworks rely on generic LVLMs which often lack emotion comprehension capabilities . In contrast, a core contribution of our work is the development of an emotion‑understanding LVLM (**Section 3.1**). We fine‑tune it using GRPO with multi‑task rewards (V‑A regression reward and emotion classification reward). This expert LVLM can evaluate emotional content more accurately and more consistently than a general‑purpose LVLM.
>
> - **The generation‑understanding‑feedback paradigm**
>
>     General LVLM-based feedback frameworks typically provide guidance on image quality or text‑image alignment, and fail to offer emotion-relevant feedback. In addition, they only support single-round generation, lacking the ability to adaptively refine prompts based on image content. EmoFeedback² introduces a generation‑understanding‑feedback paradigm to solve these limitations. Specifically, during training, we optimize the generative model with an **emotion‑aware reward feedback (RF) strategy**. During inference, we iteratively refine the input prompts using a **self‑promotion textual feedback (TF) framework**. As shown in the ablation study (**Figure 5**), RF provides fundamental emotional control, while TF enriches emotional details. This complementary capability is not available in general LVLM feedback frameworks.
>
> - **Performance comparison** with general LVLM-based feedback frameworks
>
>     To quantitatively demonstrate the advantage of our specificized LVLM and the generation‑understanding‑feedback paradigm, we additionally constructed a general LVLM-based feedback as the baseline. The baseline utilizes a general Qwen2.5‑VL‑7B‑Instruct model without emotion‑specific fine‑tuning to provide feedback. Furthermore, we also evaluated human preference reward models like ImageReward [1] and PickScore [2]. The comparative results are included in **Table R2 (Table 10 in Appendix E.1.2)**. As the general LVLM lacks emotion‑understanding capabilities, its **V‑Error (0.850) and A‑Error (0.842)** are significantly worse than those of our method **(V‑Error 0.521, A‑Error 0.710)**. While ImageReward and PickScore show improved V-A performance compared with the general LVLM, our method still outperforms them in all metrics. The results indicate that human preference may implicitly add some emotional information to the reward model. However, their performance remains inferior to the explicit emotion-aware modeling provided by our expert LVLM.
>
> **Table R2: Performance comparison across different reward models.**
> | Method | V-Error ↓ | A-Error ↓ | CLIP-Score ↑ | CLIP-IQA ↑ | Aes-Score ↑ |
> | :--- | :---: | :---: | :---: | :---: | :---: |
> | Pickscore | *0.554* | *0.788* | 25.515 | 0.790 | 5.337 |
> | ImageReward | 0.584 | 0.825 | 24.933 | 0.723 | 5.326 |
> | General | 0.850 | 0.842 | *26.776* | *0.821* | *5.399* |
> | Emotion | **0.521** | **0.710** | **26.889** | **0.880** | **5.442** |
>
> **[1]. Xu J, Liu X, Wu Y, et al. Imagereward: Learning and evaluating human preferences for text-to-image generation[J]. Advances in Neural Information Processing Systems, 2023, 36: 15903-15935.**
>
> **[2]. Kirstain Y, Polyak A, Singer U, et al. Pick-a-pic: An open dataset of user preferences for text-to-image generation[J]. Advances in neural information processing systems, 2023, 36: 36652-36663.**

---

> ### Author Response · Authors · 2025-11-27
> **Reminder for Reviewer jgo1**
>
> Dear Reviewer jgo1,
>
> We hope that our responses could adequately address your concerns. As the discussion phase deadline approaches, we warmly welcome further discussion regarding any additional concerns that you may have, and we sincerely hope you can reconsider the rating accordingly.
>
> Thank you for the time and appreciation that you have dedicated to our work.
>
> Best regards,
>
> Authors of submission 7588

---

### Author Response · Authors · 2025-12-01
**Summary to ACs**

We sincerely appreciate the extra time and considerable effort ACs have dedicated to handling this urgent situation. Your contributions are invaluable to maintaining integrity and fairness of the conference community. To support a clear and efficient evaluation of our paper and rebuttal, we have re-emphasized our key contributions and summarized the feedback from the four reviewers, along with the corresponding improvements.

- **Contributions**:

    The key challenges of C-EICG lie in achieving continuous emotion injection and high-fidelity generation. Current methods are limited by a lack of emotional feedback and poor adaptability to low-quality prompts. To overcome these limitations, we propose a novel LVLM-based generate-understand-feedback paradigm. Specifically, an emotion-aware reward feedback strategy enhances the emotional continuity of images, while a self-promotion textual feedback framework improves their emotional fidelity.

We thank all the reviewers for their encouraging evaluations and constructive feedback. We underscore **six key strengths** of our paper mentioned by the reviewers:

- **Strengths**:
    - **Novelty** of the generation-understanding-feedback reinforcement paradigm (**jgo1**, **Q4TF**, **pvHG**)
    - **Addresses** issues present in existing methods (**jgo1**, **Q4TF**)
    - Clear and good **presentation** (**jgo1**, **Zyi6**, **Q4TF**)
    - **Superior** experimental performance (**Zyi6**, **pvHG**)
    - **Comprehensive and extensive** experimental validation  (**jgo1**, **Q4TF**, **pvHG**)
    - Design philosophy **effectively integrating** emotional reasoning and feedback-based reinforcement learning (**jgo1**, **Zyi6**)

The latest revision of our paper has been uploaded, addressing all weaknesses and questions raised by the reviewers. All modifications are highlighted in **red** in the PDF. Below, we provide a **summary** of the additional experiments, explanations, and writing improvements made in response to the reviewers’ comments.

- **Additional Experiments**:

    - Added experiments and discussion on **computational efficiency**, proving that our method is comparable to other generators and user-acceptable. This paradigm innovation addresses existing limitations with minimal increase in computational cost, highlighting the application potential (**Zyi6-W5(Q4), Q4TF-W2(Q2), pvHG-W1**).
    - Added **cross-dataset** performance comparison experiments with baselines, demonstrating the generalization capability of our method (**jgo1-W2, pvHG-W2**).
    - Provided a **comparison** with general LVLM-based frameworks, showcasing our paradigm's advantage in addressing model bias (**jgo1-W4, Zyi6-W4(Q3)**).
    - Presented in-house **human verification**, demonstrating that the dataset annotations align with human perception (**Zyi6-W3(Q2), Q4TF-W1(Q1)**).
    - Included an **ablation study** on how PickScore mitigates reward hacking, along with an explanation of its integration and effectiveness (**jgo1-W3**).

- **Explanations and Discussion**:

    - Clarified the LVLM’s Emotional Evaluation **Mechanism** (**jgo1-W1**).
    - Elaborated on the **distinctions** in our dataset construction strategy (**Zyi6-W1**).
    - Explained and analyzed why Gaussian sampling **does not introduce** label noise and semantic misalignment (**Zyi6-W2(Q1)**).
    - Provided an explanation for **why** the results produced by EmoEdit and EmotiCrafter are similar (**Zyi6-W6(Q5)**).

- **Writing Improvements**:

    Provided specific experimental settings and corrected notation conventions (**pvHG-Q1, Q2, Q3**).

---

### Note · Authors · 2026-01-26

I have read and agree with the venue's withdrawal policy on behalf of myself and my co-authors.

---

### Meta-Review · Area_Chair_yL2g · 2026-01-05

**Summary:**

This paper works on continuous emotional image content generation. Authors proposed a novel generation-understanding-feedback reinforcement paradigm for C-EICG. Experimental results shows the effectiveness of the proposed method.

This paper got two 4 ratings and two 6 ratings.

The strength of this paper given by reviewers are:
1. novel paradigm design. (Reviewer jgo1, Zyi6, Q4TF, pvHG)
2. well-integrated multi-module collaboration. (Reviewer jgo1)
3. Practical Training-Free Inference Optimization. (Reviewer jgo1)
4. Comprehensive experiments. (Reviewer jgo1, Q4TF, pvHG)
5. textual feedback loop provides an intuitive and interpretable way. (Reviewer Zyi6)
6. method is effective. (Reviewer Zyi6, Q4TF)

The weakness of this paper given by reviewers are:
1. Limited Discussion on LVLM’s Emotional Evaluation Mechanism. (Reviewer jgo1)
2. Insufficient Generalization Validation. (Reviewer jgo1, pvHG)
3. Vague Explanation of Reward Hacking Mitigation. (Reviewer jgo1)
4. Lack of Comparison with LVLM-Based Alternatives. (Reviewer jgo1)
5. dataset construction strategy closely follows EmotiCrafter (Dang et al., 2025), lacks novelty. (Reviewer Zyi6)
6. problem of ground-truth Valence–Arousal annotations. (Reviewer Zyi6)
7. concerns about annotation reliability and perceptual validity on constructed dataset. (Reviewer Zyi6)
8. entire pipeline sensitive to model bias and reasoning instability. (Reviewer Zyi6)
9. computational efficiency or scalability. (Reviewer Zyi6, Q4TF, pvHG)
10. concerns on dataset construction. (Reviewer Q4TF)

Questions:
1. results produced by EmoEdit and EmotiCrafter appear highly similar. (Reviewer Zyi6)

AC carefully read authors' paper, reviewers' comments, authors' rebuttal and found authors didn't fully addressed reviewers' concerns. Details are provided below. Given these, AC decided to reject this paper.

**Reviewer Concerns:**

weakness 1. authors mentioned they mainly demonstrated the process and rationale of LVLM-based emotion assessment through explicit prompt guidance and chain-of-thought reasoning.

weakness 2. authors provided additional cross-dataset results. But we do see authors' method will be worse than FLUX in terms of CLIP-score and Aes-Score, which means authors need to sacrifice some metrics to gain V-Error or A-Error. Also it is actually interested to see the evaluation on general prompt sets to see whether the model really generalize well or not.

weakness 3. authors provided how PickScore is integrated and why pickscore. they didn't answer the reward hacking mitigation questions from reviewers. Also the reason why pickscore is not solid. There are more recent human-preference metrics like HPSv2, v3.

weakness 4. authors provides results show that their solution is better than general LVLM-based feedback frameworks.

weakness 5. reviewers mentioned about dataset constructions strategy. but authors answered different aspects from dataset construction.

weakness 6. authors addressed this problem from three perspectives: the sampling aligns with the real scenarios, the implementation strategy mitigates label noise, and the experimental results demonstrate the effectiveness.

weakness 7. authors mentioned they do incorporate human verification when constructing dataset.

weakness 8. authors provided more results to address this.

weakness 9. authors provided data points. the inference latency does increased compared with emoticrafter and sd 3.5-L.

weakness 10. authors mentioned they do have human verification for dataset construction.

**Reviewer Scores:**

Reviewer jgo1 might keep or reduce their current rating 6.

Reviewer Zyi6 might keep their current rating 4.

Reviewer Q4TF might keep or reduce their current rating 6.

Reviewer pvHG might keep their current rating 4.

---

### Decision · Program_Chairs · 2026-01-26

Reject